# The microbiome of the Arctic planktonic foraminifera *Neogloboquadrina pachyderma* is comprised of fermenting and carbohydrate-degrading bacteria and an intracellular diatom chloroplast store.

Clare Bird[1], Kate Darling[1,2], Rebecca Thiessen[3], and Anna J. Pieńkowski[4,5]

[1]Biological and Environmental Sciences, University of Stirling, Stirling, FK9 4LA, UK.
[2]School of Geosciences, Grant Institute, King's Buildings, University of Edinburgh, Edinburgh, EH9 3FE, UK.
[3]Department of Physical Sciences, MacEwan University, Edmonton, T5J 4S2, AB, Canada.
[4]Geohazards Research Unit, Institute of Geology, Adam Mickiewicz University, 61-712 Poznań, Poland
[5]Department of Arctic Geology, UNIS (University Centre in Svalbard), Longyearbyen, 9170 Svalbard, Norway

*Correspondence to*: Clare Bird (clare.bird2@stir.ac.uk)

**Abstract.** *Neogloboquadrina pachyderma* is the only true polar species of planktonic foraminifera. As a key component of the calcite flux, it plays a crucial role in the reconstruction and modelling of seasonality and environmental change within the high latitudes. The rapidly changing environment of the polar regions of the North Atlantic and Arctic Oceans poses challenging conditions for this (sub)polar species in terms of temperature, sea-ice decline, calcite saturation, ocean pH and the progressive contraction of the polar ecosystem. To model the potential future for this important high-latitude species, it is vital to investigate the modern ocean community structure throughout the annual cycle of the Arctic to understand the inter-dependencies of *N. pachyderma*. This study focusses on the summer ice-free populations in Baffin Bay. We use 16S rDNA metabarcoding and Transmission Electron Microscopy (TEM) to identify the microbial interactions of *N. pachyderma*, and PICRUSt2 to predict the metabolic pathways represented by the ASVs in the foraminiferal microbiome. We demonstrate that the *N. pachyderma* diet consists of both diatoms and bacteria. The core microbiome, defined as the 16S rDNA amplicon sequencing variants (ASVs) found in 80 % of individuals investigated, consists of six bacterial ASVs and two diatom chloroplast ASVs. On average, it accounts for nearly 50 % of the total ASVs in any individual. The metabolic pathway predictions based on bacterial ASVs suggest that the foraminiferal microbiome is comprised of monosaccharide fermenting and polysaccharide degrading bacterial species in line with those found routinely in the diatom phycosphere. On average, the two chloroplast ASVs constitute 40 % of the core microbiome and significantly, an average of 53.3 % of all ASVs in any individual are of chloroplast origin. TEM highlights the importance of diatoms to this species by revealing that intact chloroplasts remain in the foraminiferal cytoplasm in numbers strikingly comparable to the substantial quantities observed in kleptoplastic benthic foraminifera. Diatoms are the major source of kleptoplasts in benthic foraminifera and other kleptoplastic groups, but this adaptation has never been observed in a planktonic foraminifer. Further work is required to understand the association between *N.*

*pachyderma*, diatoms and their chloroplasts in the pelagic Arctic realm, but such a strategy may confer an advantage to this species for survival in this extreme habitat.

## 1 Introduction

The non-spinose planktonic foraminifera *Neogloboquadrina pachyderma* is found throughout the global ocean (Morard et al, 2024), but occurs in greatest numbers in polar, sub-polar and transitional upwelling waters. These different biogeographies and niches are reflected by the distinct *N. pachyderma* genotypes associated with the divergent ecosystems (Darling et al., 2004; 2007; 2017). For example, whilst *N. pachyderma* is the predominant planktonic foraminiferal morphospecies of the polar oceans in both hemispheres (e.g. Bé and Tolderlund, 1971, Bé, 1977), *N. pachyderma* Type I is the only genotype found in the (sub)polar North Atlantic/Arctic Ocean (Darling et al., 2004; Darling et al, 2007) and is the major marine calcifier (Kohfeld et al., 1996) in the largest ocean carbon sink in the Northern Hemisphere (Gruber et al., 2002). Present-day temperature conditions confine *N. pachyderma* Type I to the North Atlantic/Arctic Ocean water masses, where summer sea surface temperature (SST) remains below 10°C (Tolderlund and Bé, 1971; Duplessy et al., 1991). Here, *N. pachyderma* exhibits strong seasonal productivity in a highly predictable pattern. Winter mixing re-supplies nutrients to surface waters, triggering the seasonal succession of maximal phytoplankton blooms and zooplankton abundance which is followed by more nutrient depleted summer conditions (Jonkers and Kucera, 2015).

As the primary component of both the modern and Quaternary fossil (sub)polar assemblage, the calcite tests of *N. pachyderma* constitute the major contribution to seasonal and environmental change reconstructions within the North Atlantic and Arctic Ocean (e.g. Simstich, et al., 2003; Kretschmer et al, 2016; Altuna et al., 2018; Brummer et al, 2020; Livsey et al., 2020). However, the Arctic is now an unremittingly warming ecosystem, with seasonal sea ice cover constantly reducing and ice-free conditions projected to appear between 2030 and 2055 (Kim et al., 2023; Jahn et al., 2024). The North Atlantic/Arctic *N. pachyderma* is already considered to be particularly sensitive to the forecasted changes in seawater carbonate chemistry (Manno et al., 2012), with consequent implications for the calcite flux and the biological pump. Under ocean acidification conditions, Arctic *N. pachyderma* show reduced carbonate production moderated by ocean warming, making it difficult to predict future climate change impacts as the polar habitat of *N. pachyderma* shrinks. Although the North Atlantic/Arctic *N. pachyderma* population has clearly survived the extremes of Quaternary climate cyclicity in the past (Brummer et al., 2020), it is unknown whether *N. pachyderma* will find itself spatially displaced from its adaptive ecological range in the Arctic ecosystem (e.g. Jonkers et al., 2019; Greco et al., 2022), as we transition into the unknown territory of anthropogenically driven extreme global warming.

The warming ocean is affecting all marine organisms at multiple trophic levels (e.g. Poloczanska et al., 2016; Meredith et al., 2019; Deutsch et al., 2015). It has already been demonstrated that some planktonic protist species cannot track their optimal temperatures as their environment changes and may undergo extirpation once local thresholds are exceeded (Trubovitz, et al., 2020). Such thresholds are unknown for *N. pachyderma*, and there may soon be no true polar refugia into which to retreat. At

the beginning of the 21$^{st}$ century within our Baffin Bay study area, the Pikialasorsuaq (the former "North Water Polynya") was considered a region of high biological productivity (Tremblay et al., 2002; 2006). However, increasing oligotrophic conditions have been reported in the last decade driven by meltwater from the Greenland Ice Sheet and nearby glaciers. The increased stratification and reduced mixing/upwelling results in a reduction in diatom-mediated net community production (Bergeron and Tremblay, 2014). Since diatoms are considered a major food source for *N. pachyderma* (e.g. Schiebel and Hemleben 2017; Greco et al., 2021), this reduction in diatom primary productivity poses an additional challenge to *N. pachyderma* populations. To model the impending environmental consequences for this important high latitude species going forward it will be vital to investigate the modern ocean community structure throughout the annual cycle of the Arctic to understand the inter-dependencies of *N. pachyderma*.

Although our understanding of Arctic (sub)polar *N. pachyderma* annual/seasonal population structure and ecological behaviour is increasing (e.g. Carstens and Wefer, 1992; Kohfeld et al, 1996; Jonkers et al., 2010; Jonkers et al.,2013; Greco et al., 2019; Meilland et al., 2022), it is far from complete. The presence of only a single genotype (*N. pachyderma* Type I) is good news for all the Arctic ecological investigations based on this taxon (e.g. Altuna et al., 2018; Greco et al., 2019; Meilland et al., 2022) as it simplifies analyses. The microbiome, defined as the combined taxa (including food, endobionts/symbionts and parasites) within a foraminiferal specimen, has already been investigated using 16S metabarcoding of sister *Neogloboquadrina* species highlighting the diversity and complexity of the ecological community networks and symbiont/predator/prey interactions which exist between prokaryotes and protists within the water column (Bird et al., 2018). Using the 16S rDNA metabarcoding approach together with fluorescence microscopy or Transmission electron microscopy (TEM), Bird et al. (2018) determined the taxonomic character, trophic interactions, food source and putative symbiotic associations of *N. incompta* and *N. dutertrei* in the California Current system. Results highlight their similar feeding strategy of forming feeding cysts of particulate organic matter (POM) in the water column, but that such behaviour provides no clues to their choice of prey or potential symbiotic associations. Evidently, ecological concepts of individual planktonic foraminifera must be systematically revised, as each morphospecies and potentially each genotype has most likely evolved individually distinct interactions with the marine microbial assemblage. A recent study used single-cell metabarcoding targeting 18S rDNA to characterise the interactions of *N. pachyderma* with the local eukaryote community (Greco et al., 2021), since the majority of data on feeding behaviour in planktonic foraminifera suggests that biotic interactions are likely to be mainly with herbivorous eukaryotes (Kohfeld et al., 1996; Manno and Pavlov, 2014; Pados and Spielhagen, 2014; Schiebel and Hemleben, 2017). However, since no direct investigations have been carried out on the feeding behaviour or diet of Arctic *N. pachyderma*, this remains in question. The data shown in Greco et al. (2021) indicate that the *N. pachyderma* microbiome is dominated by diatoms, with Crustacea and Syndiniales (a potential parasite) also present. Here we complement this study by examining the single cell 16S rDNA metabarcodes of the *N. pachyderma* microbiome to investigate bacterial and archaeal biotic and trophic interactions and their potential symbiotic associations. In addition, we further investigate the cellular structures within *N. pachyderma* individual specimens using TEM, to examine the cellular position of the bacterial/chloroplast sources of DNA within the *N. pachyderma* cell. Our results have direct implications for understanding trophic interactions within this at-risk

habitat; for modelling future *N. pachyderma* population dynamics under climate change; and understanding the evolutionary pressures experienced by this morphospecies.

## 2 Materials and methods

### 2.1 Sampling locality and collection methods

Sampling was undertaken in Baffin Bay in July/August 2017 aboard *CCGS Amundsen* as part of ArcticNet Expedition 2017 (Leg 2b; https://arcticnet.ulaval.ca/expeditions-2017) and August/Sept. 2018 aboard the *CCGS Hudson 2018042* expedition (Fig. 1). In both cases the samples were taken in open water with no sea ice cover. Details of sampling stations and collections are listed in Table 1. Sample provenance was either individual foraminifera (Fm) or the water column (WC). Samples were analysed by 16S metabarcoding of the foraminiferal microbiome or the water column bacterial assemblages from stations along the *Amundsen* cruise track.

Foraminifera were collected at seven stations by vertical net tow from 200 m depth to the surface. Foraminifera for genotyping and microbiome analysis were wet picked on board, rinsed in 0.2 μm filtered surface seawater and preserved in 100 μl RNALater® (Ambion™). Samples were stored at 4°C for 4 hours and then transferred to a -20°C freezer until processing. CTD data and water samples were collected from three stations (101, 115 and 323) in northern Baffin Bay (Fig. 1; Table 1) in 2017. Stations 101 and 115 are both situated within the biologically important Pikialasorsuaq between Greenland and Canada, which remains sea-ice-free in winter (Eegeesiak et al., 2017). Station 101 is located close to southeast Ellesmere Island in relatively shallow water (350 m depth). Station 115 is at a similar latitude to Station 101 but deeper at 653 m and closer to Greenland. Station 323 is situated further south and outside the Pikialasorsuaq in Lancaster Sound at the entrance to the Northwest Passage. It is the deepest of the three stations at 789 m.

Water samples were collected from five depths: surface, 50 m, 100 m, 150 m, and 200 m. 2L of seawater was filtered from each depth at each station on to 0.2 μm polycarbonate filters. Filters were then individually placed in a 1.5ml microfuge tube and covered with RNA*Later*® (Ambion™). Tubes were stored at 4°C for 4 hours and then transferred to a -20°C freezer until processing. Foraminifera for TEM analysis were collected on the CCGS Hudson cruise (2018042) in 2018 (Table 1). They were wet picked as described above and placed directly in TEM buffer (4 % glutaraldehyde, 2 % paraformaldehyde in salt-adjusted phosphate buffered saline (PBS with 24.62g/L NaCl added)), stored at room temperature for 12 hours, then kept at 4°C until further processing.

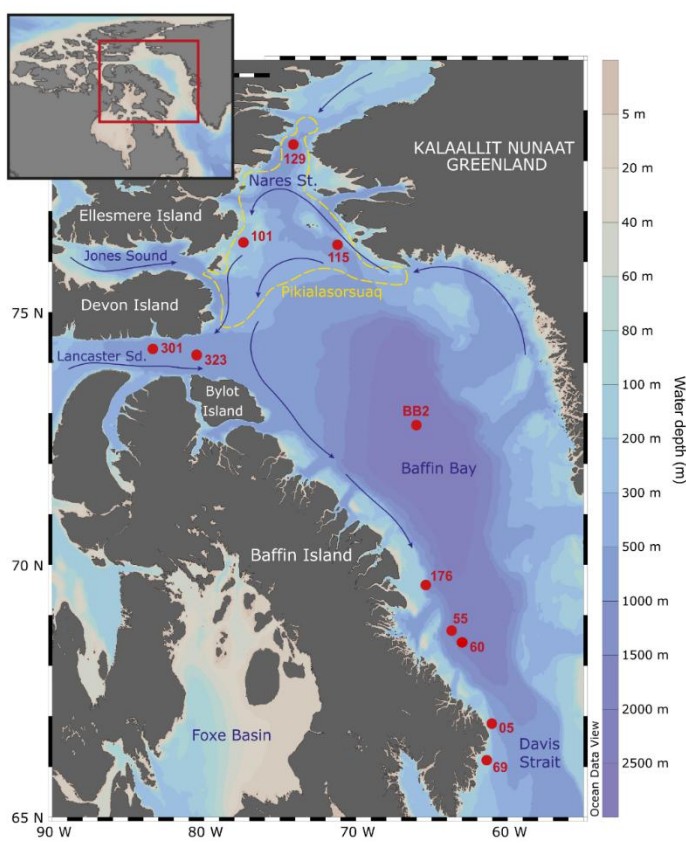

Figure 1. Map of sampling stations (numbered red spots) and the site of Pikialasorsuaq (the former 'North Water Polynya' (dashed yellow line); Eegeesiak et al., 2017). Stations BB2, 101, 115, 129, 301 and 323 were part of the *CCGS Amundsen ArcticNet* Expedition 2017 and Stations 05, 55, 60 and 69 were part of the *CCGS Hudson 2018042* Expedition 2018. Inset shows the sampling location within the wider region. The base maps were drawn in Ocean Data View v.5.6.2 (Schlitzer 2018).

Table 1. Stations and sample information including provenance, depth, and analysis type. Specimen IDs are either WC (water column) or Fm (foraminifera). This is followed by station identification (e.g. 101), and water depth (e.g. 050 = 50 metres), and replicate ID (e.g. a, b or c etc.)

| Cruise | Station | Sampling date | Specimen IDs | Latitude (°) | Longitude (°) | Water depth (m) | Sample depth (m) | Provenance | Analysis |
|---|---|---|---|---|---|---|---|---|---|
| AMD2017-2B | 101 | JUL-24-2017 | WC101_000a | 76.3844 | -77.4033 | 350 | surface | Water column | microbiome |
|  |  |  | WC101_050a | 76.3844 | -77.4033 | 350 | 50 | Water column | microbiome |

| | | WC101_050b | | | | | | |
|---|---|---|---|---|---|---|---|---|
| | | WC101_050c | | | | | | |
| | | WC101_100b | 76.3844 | -77.4033 | 350 | 100 | Water column | microbiome |
| | | WC101_150a | 76.3844 | -77.4033 | 350 | 150 | Water column | microbiome |
| | | WC101_150b | | | | | | |
| | | WC101_200a | 76.3844 | -77.4033 | 350 | 200 | Water column | microbiome |
| | | WC101_200b | | | | | | |
| 101 | JUL-24-2017 | Fm101a | 76.3844 | -77.4033 | 350 | surface-200m | Foraminifera | microbiome, genotyping |
| | | Fm101b | | | | | | |
| | | Fm101c | | | | | | |
| | | Fm101d | | | | | | |
| | | Fm101e | | | | | | |
| | | Fm101f | | | | | | |
| | | Fm101g | | | | | | |
| 115 | JUL-26-2017 | WC115_000a | 76.3419 | -71.2192 | 653 | surface | Water column | microbiome |
| | | WC115_000b | | | | | | |
| | | WC115_050a | 76.3419 | -71.2192 | 653 | 50 | Water column | microbiome |
| | | WC115_050b | | | | | | |
| | | WC115_100a | 76.3419 | -71.2192 | 653 | 100 | Water column | microbiome |
| | | WC115_100b | | | | | | |
| | | WC115_150a | 76.3419 | -71.2192 | 653 | 150 | Water column | microbiome |
| | | WC115_150b | | | | | | |
| | | WC115_200a | 76.3419 | -71.2192 | 653 | 200 | Water column | microbiome |
| | | WC115_200b | | | | | | |
| | | Fm115a | 76.3419 | -71.2192 | 653 | surface-200m | Foraminifera | microbiome, genotyping |
| | | Fm115b | | | | | | |
| 323 | JUL-31-2017 | WC323_000a | 74.1593 | -80.4753 | 789 | surface | Water column | microbiome |
| | | WC323_050a | 74.1593 | -80.4753 | 789 | 50 | Water column | microbiome |
| | | WC323_050b | | | | | | |
| | | WC323_100a | 74.1593 | -80.4753 | 789 | 100 | Water column | microbiome |
| | | WC323_100b | | | | | | |
| | | WC323_150a | 74.1593 | -80.4753 | 789 | 150 | Water column | microbiome |
| | | WC323_150b | | | | | | |
| | | WC323_200a | 74.1593 | -80.4753 | 789 | 200 | Water column | microbiome |
| | | WC323_200ab | | | | | | |
| 323 | JUL-31-2017 | Fm323a | 74.1593 | -80.4753 | 789 | surface-200m | Foraminifera | microbiome, genotyping |
| | | Fm323b | | | | | | |
| 176 | JUL-21-2017 | Fm176a | 69.6032 | -65.3938 | 281 | surface-200m | Foraminifera | microbiome, genotyping |
| | | Fm176b | | | | | | |
| | | Fm176c | | | | | | |
| | | Fm176d | | | | | | |

| | | | | | | | | | |
|---|---|---|---|---|---|---|---|---|---|
| | BB2 | JUL-22-2017 | FmBB2a FmBB2b FmBB2c FmBB2d FmBB2e | 72.7678 | -66.0002 | 2372 | surface-200m | Foraminifera | microbiome, genotyping |
| | 129 | JUL-29-2017 | Fm129a Fm129b | 78.3254 | -74.1124 | 514 | surface-200m | Foraminifera | microbiome, genotyping |
| | 301 | AUG-03-2017 | Fm301a Fm301b Fm301c Fm301d Fm301e Fm301f | 74.2778 | -83.3641 | 716 | surface-200m | Foraminifera | microbiome, genotyping |
| Hudson 2018042 | 05 | AUG-23-2018 | BB1 | 66.8605 | -61.0668 | 337 | 100 | Foraminifera | TEM |
| | 05 | AUG-23-2018 | BB2 | 66.8605 | -61.0668 | 337 | 100 | Foraminifera | TEM |
| | 55 | AUG-31-2018 | BB8 | 68.6999 | -63.7084 | 1560 | 50 | Foraminifera | TEM |
| | 60 | SEP-02-2018 | BB9B | 68.543415 | -63.461252 | 1543 | 100 | Foraminifera | TEM |
| | 60 | SEP-02-2018 | BB9C | 68.543415 | -63.461252 | 1543 | 100 | Foraminifera | TEM |
| | 69 | SEP-04-2018 | BB11 | 66.1371 | -61.3659 | 160 | 100 | Foraminifera | TEM |
| | 69 | SEP-04-2018 | BB12 | 66.1371 | -61.3659 | 160 | 100 | Foraminifera | TEM |

## 2.2 DNA extractions, foraminiferal 18S rDNA genotyping and 16S rDNA metabarcoding.

Downstream washing of individual cells preserved in RNALater® for genotyping was carried out to remove the test and test-associated external contaminants according to Bird et al. (2017). DNA was extracted from individual foraminifera in 40µl DOC buffer (Holzmann and Pawlowski, 1996) to identify the specific genotype. PCR amplification of the foraminiferal 18S

rDNA gene was performed with three rounds of PCR using a Phire Hot start DNA polymerase master mix (Thermo-Scientific), 3 % DMSO and an annealing temperature of 58°C with 25 cycles. DNA was diluted 1 in 20 in PCR grade water. Primer pairs were as follows: Primary PCR: C5-sB, secondary PCR: N5-N6, tertiary PCR: 14F1-N6. PCR products between rounds were diluted 1 in 100 PCR grade water and 1 µl was used in the following round of PCR. Cloning to account for intra-individual variation was carried out according to Darling et al. (2016). DNA sequencing was carried out using the BigDye® Terminator

v3.1 Cycle Sequencing Kit and an ABI 3730 DNA sequencer (both Applied Biosystems). Filtrate from water samples was extracted for DNA using the DNeasy power water kit (Qiagen). Filters were removed from RNA*Later*salt a (Ambion™), placed in clean 1.5 ml microfuge tubes and centrifuged for 1 min at 10,000 xg. Excess RNA*Later*® (Ambion™) was removed and the filter was transferred to the bead beating tubes of the DNeasy power water kit and processed following the manufacturer's protocol. A control DNeasy power water kit extraction was carried out in parallel using a clean filter. In addition

to the foraminiferal and water sample processing, four reagent controls were also processed. These were composed of two

PCR controls containing no DNA template, an extraction control containing 2.5µl DOC buffer only and an extraction control containing 1µl of elute from a Qiagen DNA extraction of a clean 0.2 µm polycarbonate filter.

PCR was used to amplify the V4 region of the 16S rDNA gene of bacteria and chloroplasts. PCR reactions using 515F forward (Parada et al., 2016) and 806R reverse (Apprill et al., 2015) primer pair modified from the original primer pair (Caporaso et al., 2011, Walters et al., 2016) were performed in triplicate. Each reaction contained 1 Unit Phusion DNA polymerase (ThermoScientific), 1 x Phusion HF buffer, 0.2 mM each dNTP, 0.4 µM of each primer, 0.4mM $MgCl_2$ and 2.5 µl (foraminifera) or 1 µl (water column) of template DNA in a 50 µl volume made up with PCR grade water (Sigma). All PCR reactions were set up in a UV sterilization cabinet (GE healthcare). Reaction tubes and PCR mixtures were treated for 15 minutes with 15 W UV light (wavelength = 254 nm) to destroy contaminating DNA, prior to addition of dNTPs, DNA polymerase primers and template DNA (Padua et al., 1999). Triplicate PCR reactions were pooled before purification with the Wizard® SV Gel and PCR Clean–Up System (Promega). The purified amplicons were quantified using a Qubit® 2 fluorometer (ThermoFisher Scientific) prior to pooling at equimolar concentrations for DNA sequencing. DNA sequencing was performed at Edinburgh Genomics using an Illumina MiSeq v3 to generate 253 base pair (bp) paired-end reads.

### 2.3 Quality filtering paired end reads, rarefaction, taxonomic assignment and sequence filtering.

The Quantitative Insights in Microbial Ecology 2 pipeline (QIIME2, Bolyen et al., 2019) was used for initial analyses. Sequences were trimmed and denoising was carried out using the DADA2 plugin (Callahan et al., 2016) for quality filtering, dereplication, removal of singletons, chimera identification and removal and merging paired-end reads. This method generates amplicon sequence variants (ASVs) and a set of representative sequences. Alpha rarefaction was carried out in QIIME2 (metrics: observed OTUs; Shannon; and Faith PD). Samples were rarefied to the lowest sequencing depth observed across all samples (25,064) and sampling depth was adequate across all samples (Fig. A1). A total of 60 samples including 28 foraminiferal samples, 28 water samples and 4 negative controls, produced a total of 4290 ASVs across 5,802,211 counts. Amplicon sequencing variants with a total frequency count of 50 or less across all 60 samples were removed from the sample set, leaving 1717 ASVs. Taxonomy was assigned using an SKlearn classifier pre-trained on the database SILVA-132 99 % OTUs from the V4 515f/806R region of the sequences (Quast et al., 2013). Taxonomic-based filtering was then carried out to remove ASVs assigned to mitochondria, eukaryotes, and those not assigned beyond Kingdom level, and contaminant removal (26 identified ASVs) was carried out in the R package *Decontam* (Davis et al., 2018) in R v 4.0 (R Core Team, 2017) using the prevalence with batch methods. After filtering, 1548 ASVs remained. For plotting and comparisons, the read counts for each ASV in each sample were converted to percentage reads, i.e., relative abundances. Therefore, all percentages presented are the relative abundances of individual ASVs or groups of ASVs being discussed.

### 2.4 Statistical analyses

Absolute count data were transformed to centred log-ratios suitable for statistical analysis of compositional data using q2-Gemelli (Martino et al., 2021). Robust Aitchison distances were calculated (Martino et al., 2019) and a PCA was performed

in Vegan v2.6-10 (Oksanen et al., 2017) in R (v 4.0 and 4.4.3; R Core Team, 2017). This was visualised with ggplot2 (Wickham, 2009).Statistical analyses were also performed in Vegan v2.6-10. To determine if Provenance, sample Depth or
Station significantly affected the assemblages *Adonis2* was used to perform PERMANOVA (default 999 permutations). Pairwise PERMANOVA tests were performed using the pairwise.adonis2 function (pairwiseAdonis v 0.4.1; Martinez Arbizu, P., 2020) with 999 permutations. To account for multiple testing across pairwise comparisons, raw P-values were corrected using the Benjamini-Hochberg method (implemented via p.adjust in base R), and comparisons with adjusted P-values (q-values) < 0.05 were considered statistically significant. Where sample numbers were small and unequal, a dispersion test
(implemented with betadisper()) was carried out on the Aitchison dissimilarity matrices to test group dispersion homogeneity, since PERMANOVA is sensitive to unequal sample numbers, particularly when group dispersions differ. To assess differential microbial composition between stations in subsets where PERMANOVA was inappropriate a Wilcoxon rank-sum test was implemented in ALDEx2 v1.38.0 (Analysis of Differential Abundance Taking Sample and Scale Variation into Account; Fernandes et al., 2014). ALDEx2 models high-throughput compositional data using centred log-ratio (clr) transformation and
Monte Carlo sampling (set to 1000, and denominator = all) from a Dirichlet distribution, which accounts for within-group variance, and is robust to small sample sizes and differences in dispersion.

    To investigate differential abundance in the taxa associated with the different provenances, the packages Phyloseq v1.50.0 (McMurdie and Holmes, 2013) and DESseq2 v1.46.0 (Love at al., 2014) were used. The package DESeq2 was used rather than ANCOM because ANCOM assumes that <25 % of the ASVs are changing between provenances, and here this assumption
does not hold true (Mandal et al., 2015). Compositional differences and specific taxa that were significantly different between provenances were identified using log2 of fold change analysis in DESeq2 by converting the phyloseq-object, containing the raw frequency counts, to a DESeq2 object. The DESeq2 analysis was run with size factor type set to "poscounts" which allows values of zero in the sample counts and accounts for the data transferal from a phyloseq-object (van den Berge et al., 2018). The significance test was set to "Wald", and a "local" fit type for fitting of dispersions.

The core microbiome, here defined as ASVs present across 80 % of the foraminifera, was identified using *Microbiome* (Lahti and Shetty et al., 2017).

    Functional predictions of the foraminiferal microbiome compared to the wider water column assemblage were made using PICRUSt2 (Phylogenetic Investigation of Communities by Reconstruction of Unobserved States, Douglas et al., 2020). The inputs to PICRUSt2 were the representative sequences fasta file and the ASV frequency table (converted to biom format)
generated in QIIME2. The default full pipeline was run with the addition of "--per_sequence_contrib" and "--coverage" to give copy number normalised, community wide pathway abundances to compare between the provenances. Evolutionary Placement Algorithm -Next Generation (EPA-NG) phylogenetic placement of reads (Barbera et al., 2019) was used with the default cutoff nearest sequence taxon index (NSTI) of 2.0 which removed 8 of 1548 ASVs from downstream analysis, as these could not be satisfactorily placed in the tree. ALDEx2 (Fernandes et al., 2014) was also used to assess the differential
abundance of functional pathway predictions between the foraminiferal microbiome and the water column assemblage. PICRUSt2 generated pathway abundance data was rounded to integers then input to aldex.clr which generates centred log-

ratio transformed values (number of Monte-Carlo instances = 1000, and denominator = iqlr). The Welch's t test (aldex.ttest) and an estimate of the effect size and the within-and-between Provenance values (aldex.effect) were calculated from the output of aldex.clr. The dataset was then filtered for pathways that were significantly differentially abundant between Provenances (Benjamini-Hochberg corrected P value <0.05 and effect size >1).

## 2.5 Transmission electron microscopy

TEM was used to observe and document the structural relationships between any endobiotic micro-organisms and foraminiferal cells. After fixation in TEM fixative (see methods section 2.1), specimens were post–fixed in 1 % Osmium Tetroxide in 0.1 M Sodium Cacodylate for 45 minutes, followed by a further three 10-minute washes in distilled water. Specimens were then set in small cubes of 1 % low melting point agarose and decalcified in 0.1 M EDTA (pH 7.4) for 1 hour and 48 hours at 4 °C. Fixed cells were then dehydrated in 50 %, 70 % and 90 % ethanol for 2 x 15 minutes followed by 100 % ethanol 4 x 15 minutes. Two 10–minute changes in Propylene Oxide were carried out prior to being embedded in TAAB 812 resin. Sections, 1 μm thick, were cut on a Leica Ultracut ultramicrotome, stained with Toluidine Blue, and then viewed under a light microscope to select suitable specimen areas for investigation. Ultrathin sections, 60 nm thick, were cut from selected areas, stained in Uranyl Acetate and Lead Citrate and then viewed with a JEOL JEM–1400 Plus TEM. Both osmium tetroxide and uranyl acetate used here bind to unsaturated lipids such that they appear dark in TEM imaging.

## 3 Results

### 3.1 The water column

Water samples were taken from three locations: Stations 101, 115 and 323 (Fig. 1, Table 1). At Station 115 to the east of the Pikialasorsuaq, the water is derived from warm Atlantic water (e.g. Melling et al., 2001; Vincent, 2019) with SSTs as high as 4°C accompanied by a steep thermocline to 40 m (Fig. 2). At Stations 101 and 323 (west and southwest of the Pikialasorsuaq), the upper level water temperatures were colder due to Pacific-derived water entering via the colder Arctic Ocean (Tremblay et al., 2002; Bergeron and Tremblay, 2014). Whilst Station 323 had a surface temperature of 3°C, it very rapidly dropped to -1°C by 20 m and Station 101 had a surface maximum temperature of only 0.75°C. The chlorophyll maximum was closely associated with the temperature maximum at all stations.

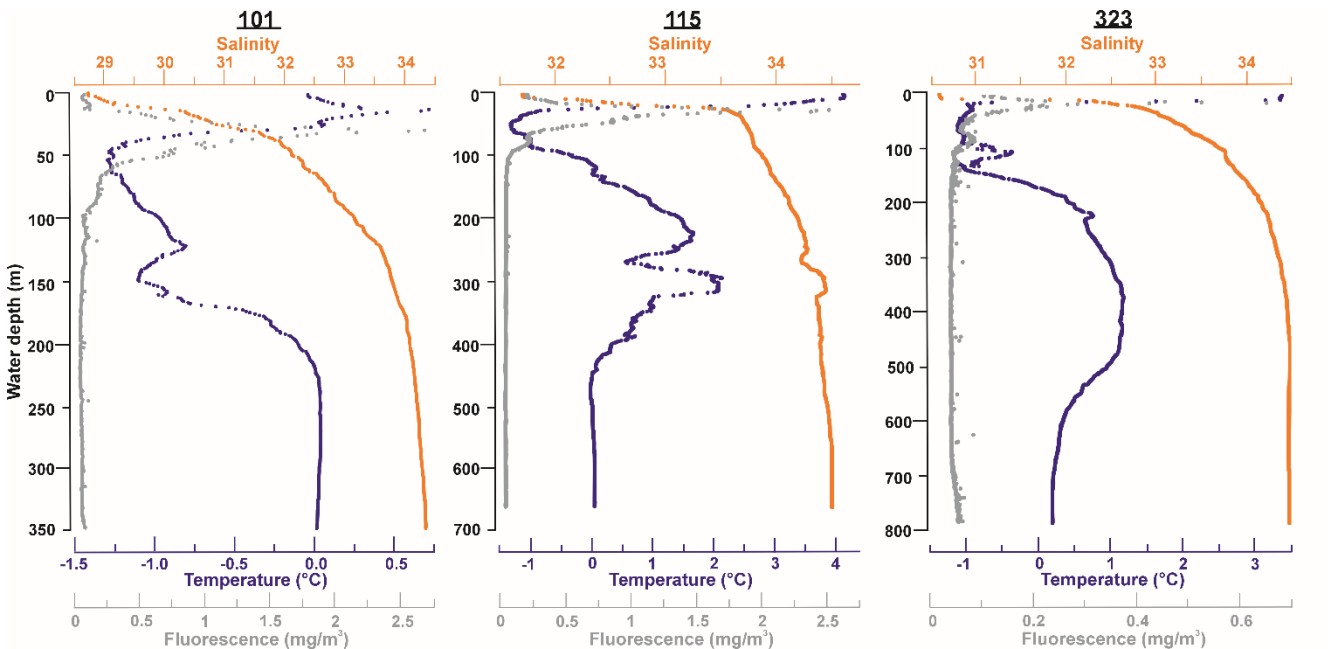

Figure 2. CTD plots for temperature, salinity, and fluorescence at Stations 101 (350m), 115 (653m) and 323 (789m) where both water and foraminifera were collected.

The general microbial assemblages in the water column across the three stations (101, 115 and 323) displayed a similar pattern of assemblage composition with depth (Fig. 3). Surface waters contained either no, or extremely low relative abundances of chloroplast or archaeal ASVs. Chloroplast ASV relative abundance increased steeply with depth however, with the highest abundance found in the 50 m water samples, before numbers reduced again. This pattern agrees with our CTD data, where the
chlorophyll maximum occurred between 20-40 m across all stations (Fig. 2). The chloroplast ASVs made up an average of 3.51 % of the ASVs in the water column. Archaeal ASVs (candidate Phylum Thermoplasmatota, and the Crenarchaeota) are most predominant below 50 m, peaking in the 100-150 m water samples.

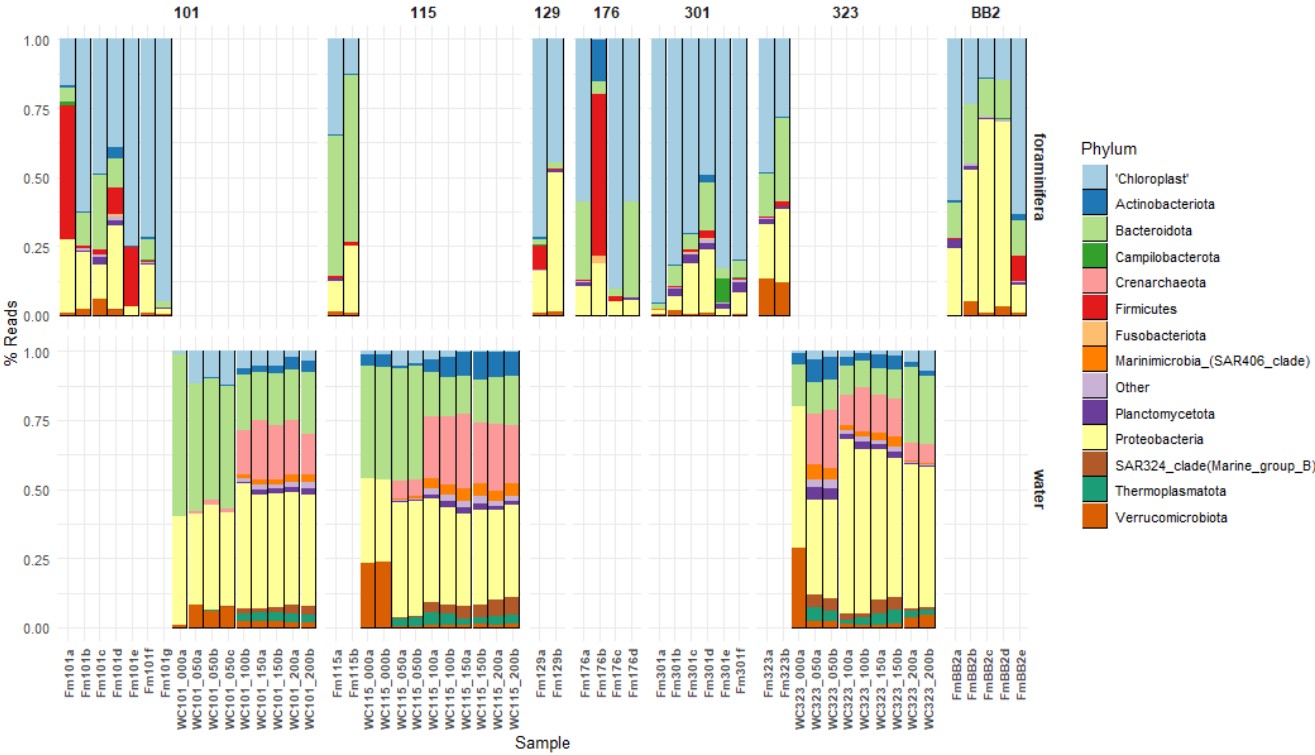

Figure 3. The relative abundance of 16S rDNA ASVs generated from the foraminiferal specimens (Fm) and the water column (WC). Note that foraminifera (top row) were successfully processed from 7 stations (see Fig. 1) and water was collected and processed from three stations (bottom row). Taxa are shown at the phylum level except for chloroplast derived 16S ASVs, which are grouped together. Sample IDs (Table 1) are found on the x-axis.

## 3.2 Statistical comparison of water column and foraminiferal ASVs

The combined microbial assemblages of the 28 water column samples and the 28 foraminifera specimens (see Table 1) consisted of 1548 identified ASVs after filtering. By far the most prevalent ASVs in the foraminifera are those from chloroplasts (averaging 53.3 % relative abundance), and particularly diatom chloroplasts (averaging 44.5%). In the water column, Proteobacteria ASVs are the most abundant (averaging 41.2 %, Fig. 3).

### 3.2.1 Station and Depth as factors influencing microbial assemblages

To assess the influence of environmental factors on microbial assemblages in the water column, the effects of Station and Depth were tested across the three stations with multiple depth samples. PERMANOVA (Table A1) revealed that Depth was a significant driver of community composition, explaining 55 % of the variation (Pr = 0.001). In contrast, Station explained

only 6.9% and was not significant, either globally (Pr = 0.458) or in pairwise comparisons (see Pairwise PERMANOVA in Supplementary Material). PCA ordination (Fig. A2) supported this, showing clearer clustering by Depth than by Station.

To control for vertical variation, a subset of 50 m water samples was analysed. Station explained 98% of variation and was significant in the global PERMANOVA (Pr = 0.012); however, none of the pairwise comparisons were significant, and low sample sizes (n = 2–4) limit interpretation.

In contrast, for foraminiferal samples, Station explained 48.3 % of the variation and was significant in the global PERMANOVA (Pr = 0.003, Table A1). Pairwise comparisons (see Supplementary Material) revealed significant differences in four of 21 station pairs (after correction), indicating some degree of spatial structuring, and this is reflected in the PCA ordination of the foraminifera samples (Fig. A3). Specifically, the foraminiferal microbiome at Station 301 differed significantly from those at Stations BB2, 176, and 101. However, samples from Stations 101, 115, and 323, where water column samples were also collected, did not differ significantly, supporting the water column result of minimal microbial differentiation among these three stations. To improve statistical power, the foraminiferal analysis was repeated using only the four best-replicated stations (Stations 101, 176, BB2, and 301; n ≥ 4). In this dataset, Station remained significant, explaining 41 % of the variation (Pr = 0.003), and five of six pairwise comparisons were also significant. These included all previously identified station differences, plus BB2 versus 101.

To enable a direct comparison with the water column dataset, a subset (F3) of foraminiferal samples was created using only those collected at the three stations where water column samples were also available (Stations 101, 115, and 323). This geographically matched subset was used to test further whether the stronger Station effect observed in the full foraminiferal dataset simply reflected broader spatial coverage. Due to limited replication (n = 7, 2, and 2) and heterogeneous dispersion (Table A1), PERMANOVA was not appropriate. Instead, ALDEx2 with pairwise Wilcoxon tests identified only one ASV with significantly different abundance between any station pair, suggesting minimal spatial structuring in this subset, consistent with the water column results.

To evaluate whether the observed station effect was maintained across both Provenances within a geographically consistent subset, a combined dataset (FW) was created, including all foraminiferal and water column samples from the three co-sampled stations (101, 115, and 323). Testing the effect of Station in this FW dataset revealed no significant differences (Pr = 0.423), with Station explaining just 5.4 % of the variation. PCA ordination of the FW dataset (Fig. A4) similarly showed stronger separation by Provenance than by Station, suggesting that sample type has a greater influence on community composition. Together, these results indicate that microbial community composition varies little among these three stations, regardless of sample Provenance.

**3.2.2 Multivariate analyses of foraminiferal versus water column ASV composition.**

Multivariate analyses were conducted on two datasets to compare microbial community composition between Provenances (foraminifera vs. water column). The first, dataset FW, included all water column and foraminiferal samples from the three

co-sampled stations (n = 28 water; n = 11 foraminifera). The second, dataset 101, comprised only samples from Station 101, allowing Provenance to be assessed independently of Station (n = 9 water; n = 7 foraminifera).

PERMANOVA on the FW dataset revealed a significant difference in ASV composition between Provenances, which explained 9.7% of the variation (Pr = 0.013; Table A1). PCA ordination supported this, showing clearer separation by Provenance than by Station (Fig. A4).

Analysis of dataset 101 reinforced this result where Provenance explained 41.7% of the variation (Pr = 0.002; Table A1), and PCA ordination showed distinct clustering of foraminiferal samples apart from water column samples (Fig. A5). This indicates

that, even within a single water column, microbial communities associated with foraminifera differ significantly from those in surrounding water.

However, although PCA ordinations supported the PERMANOVA results, tests for homogeneity of multivariate dispersion indicated significant heterogeneity in dispersion between Provenances in both datasets (Table A1), which can violate PERMANOVA assumptions and complicate interpretation of compositional differences. To address this and identify ASVs

potentially driving these compositional differences between Provenances, differential abundance analyses were conducted on the ASVs across both datasets.

### 3.2.3 Differential abundance analysis of ASVs in foraminifera versus the water column

Deseq2 fold change analysis of the FW dataset identified that 572 of 1207 ASVs are driving the significant compositional differences between Provenances (P<0.05). All but 13 of those 572 ASVs are significantly more abundant in the water column.

These include three Firmicutes (ASVs 1402, 609, 391), three Gammaproteobacteria (ASVs 927, 420, 116), two Bacteroidota (ASVs 743, 46), two Actinobacteriota (ASVs 1403, 509), one chloroplast ASV of unknown origin (ASV 1538) and two chloroplast ASVs from Class Bacillariophyceae (ASV355, *Fragilariopsis cylindricus*, and ASV 956, *Chaetoceros gelidus*) (Table A2, Fig. 4).

Deseq2 fold change analysis of the 101 dataset identified 393 of 1013 ASVs that are driving the significant compositional

differences between Provenances (P<0.05). All but 14 of those are significantly more abundant in the water column. These include three Firmicutes (ASVs 1402, 609, 391) five Gammaproteobacteria (ASVs 927, 194, 149, 133, 116), two Bacteroidota (ASVs 743, 46) one Actinobacteriota (ASV509), one chloroplast ASV of unknown origin (ASV472), and two chloroplast ASVs from Class Bacillariophyceae (ASV355, *Fragilariopsis cylindricus*, and ASV 956, *Chaetoceros gelidus;* Table A2, Fig. A6). Importantly, ten of the ASVs that are significantly more abundant in the foraminifera are supported in both datasets.


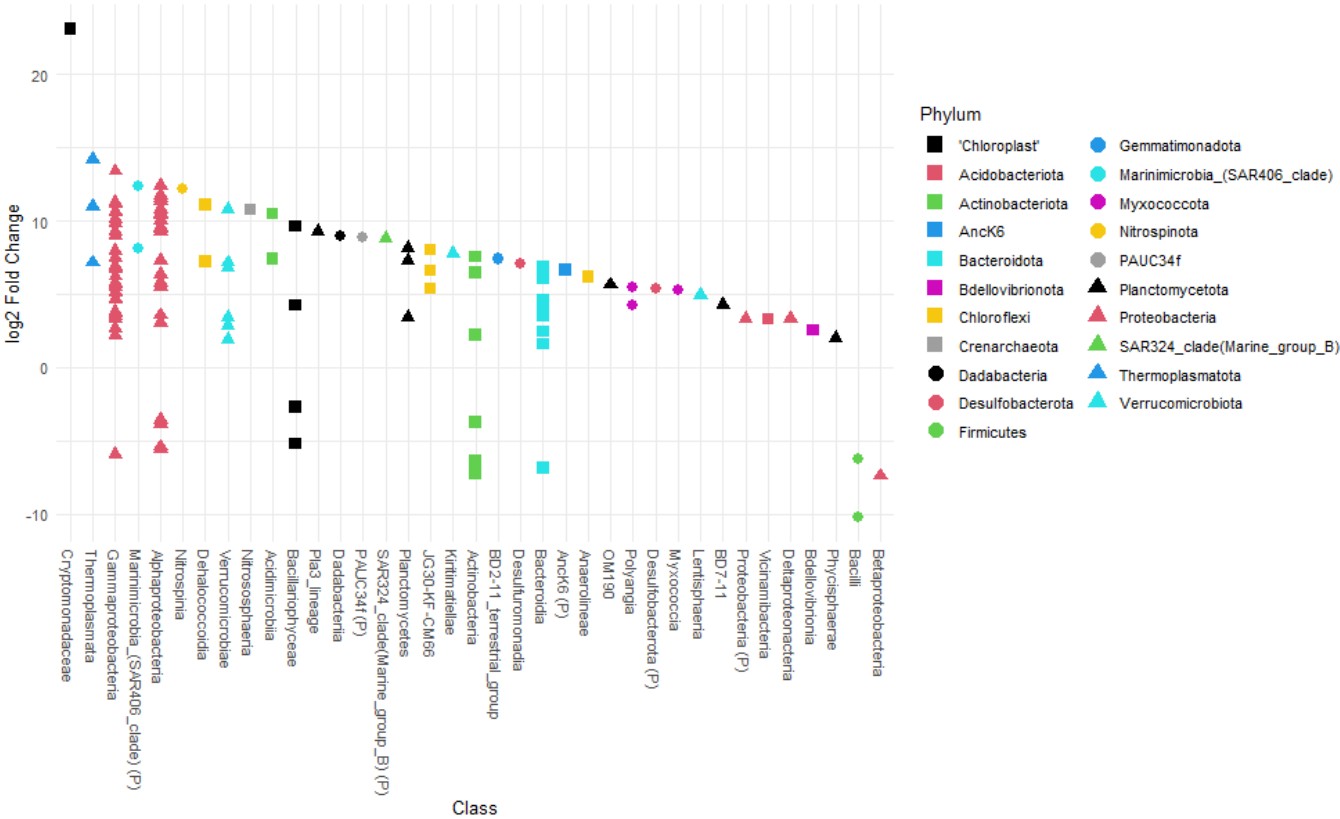

Figure 4. Differential abundance testing of FW dataset ASVs between Provenances using *DESeq2*. The Log2 fold change in ASVs is the log-ratio of the ASV means in the water column and foraminifera. ASVs with positive Log2 fold change are significantly more abundant in the water column assemblage and ASVs with negative values indicate ASVs which are significantly more abundant in the foraminiferal assemblages. The Class, or the highest level of taxonomic assignment available for each ASV, is given on the x-axis. (P) =Phylum

### 3.2.4 Differential abundance analysis of predicted functional pathways in the foraminiferal microbiome versus the water column assemblage.

PICRUST2 was used to identify possible functional pathways (using the MetaCyc database, Caspi et al., 2014) and to calculate functional pathway abundances based on estimated abundances of gene families that can be linked to reactions within those pathways. 415 pathways were identified, and their differential abundances were compared between the microbiome of *N. pachyderma* and the water column assemblage using ALDEx2. Ninety-two pathways were identified as significantly differentially abundant between the two Provenances (effect size >1 and Benjamini-Hochberg adjusted P<0.05). These 92 pathways were grouped according to metabolic types to identify broader metabolic processes that were significantly different within the two Provenances (Fig. 5). Of the 92 pathways, 38 were significantly more abundant in the foraminiferal microbiome.

They include L-lysine biosynthesis (PWY-2941), peptidoglycan synthesis and recycling (four pathways, 897 ASVs), carbohydrate degradation (nine pathways, 624 ASVs), fermentation (four pathways, 496 ASVs) and the production of the secondary metabolite palmitate (PWY-1479, 4 ASVs) and butanediol production (two pathways, 49 ASVs, Fig. 5). Table A2 identifies those pathways identified in the significantly differentially abundant ASVs. The remaining 54 pathways were significantly more abundant in the water column assemblage, including 14 pathways involved in aromatic compound degradation, pathways for co-factor carrier and vitamin biosynthesis, inorganic nutrient metabolism, nucleoside and nucleotide biosynthesis, fatty acid and lipid biosynthesis, and C1 compound utilisation.



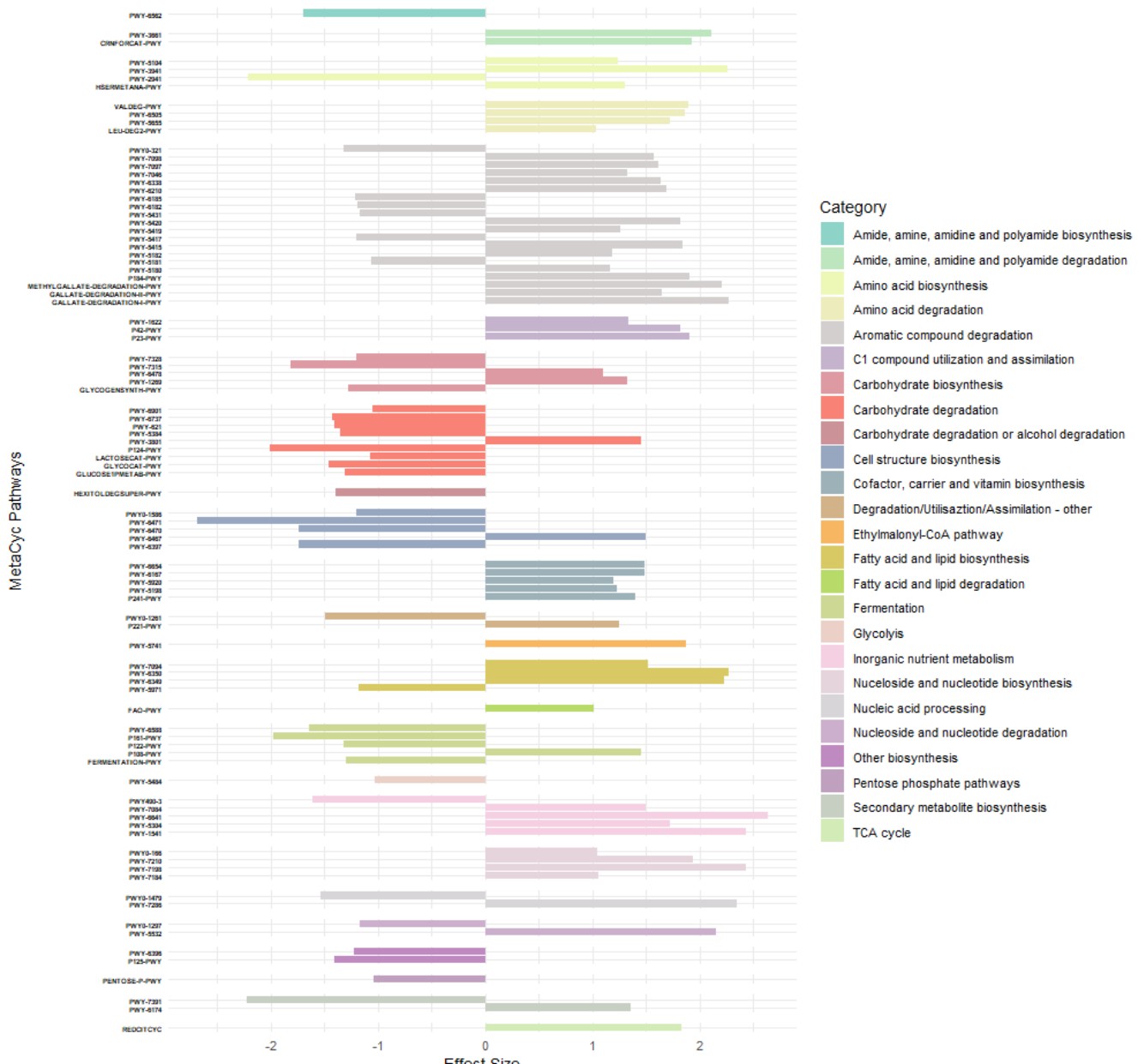

Figure 5. MetaCyc Pathways identified as being significantly differentially abundant between the two Provenances are shown (y-axis) with their 'Effect Size' indicated by the x-axis. Negative values indicate the pathway is more abundant in the foraminiferal microbiome and positive values indicate the pathway is more abundant in the water column assemblage. Pathways are grouped by broader metabolic categories identified in the key.

### 3.3 Foraminiferal ASV profiles

All the foraminifera were genotyped as *N. pachyderma* Type I (NCBI GenBank Accession numbers OR137988-OR138014), consistent with it being the only genotype found in the Arctic region to date (Darling et al., 2004; 2007), indicating that variation in ASV composition cannot be driven by genotype differences..

### 3.3.1 Bacterial ASV profiles in *N. pachyderma*

Amplicon sequencing variants were assigned to 1367 distinct bacterial and archaeal taxa across all water and foraminifera samples within 29 Phyla and 60 Classes. The major groups were Class Gammaproteobacteria which contributed on average 18.8 % of ASVs, Phylum Bacteroidota 14.6 %, and Class Bacilli (Phylum Firmicutes) 5.1 %. The only other groups that contributed >1 % of ASVs are Phylum Actinobacteria, Class Alphaproteobacteria and Class Verrucomicrobiae. The other 545 classes all contribute < 1 % each to the ASV total. This distribution reflected the ASVs driving significant compositional differences between the Provenances (ASVs from Class Gammaproteobacteria, Phylum Bacteroidota, Phylum Firmicutes and Phylum Actinobacteria. Section 3.3, Table A2).

### 3.3.2 Chloroplast ASV profiles in *N. pachyderma*

Amplicon sequencing variants  are assigned to 181 distinct chloroplast ASVs corresponding to 14 unique chloroplast-containing classes across all water and foraminifera samples, contributing, on average, 53.3 % of all ASVs in the foraminifera and only 3.51 % in the water column (Fig. 3).  More specifically, those ASVs assigned to diatom chloroplasts (Class Bacillariophyceae) contributed on average to 44.5 % of all ASVs in the foraminifera and only 2.36 % in the water column, highlighting the major importance of diatoms in the diet of the foraminifera compared to other phytoplankton taxa. Three chloroplast ASVs drive the significant difference between the foraminifera and the water column, due to greater abundance in the foraminifera. These are one ASV (ASV1538) identified only as "Chloroplast", and two diatom ASVs identified as *Fragilariopsis cylindicus*. (ASV355) and *Chaetoceros gelidus* (ASV956) (Table A3). The relative abundance of chloroplast ASVs in each sample is shown in Fig. 6.

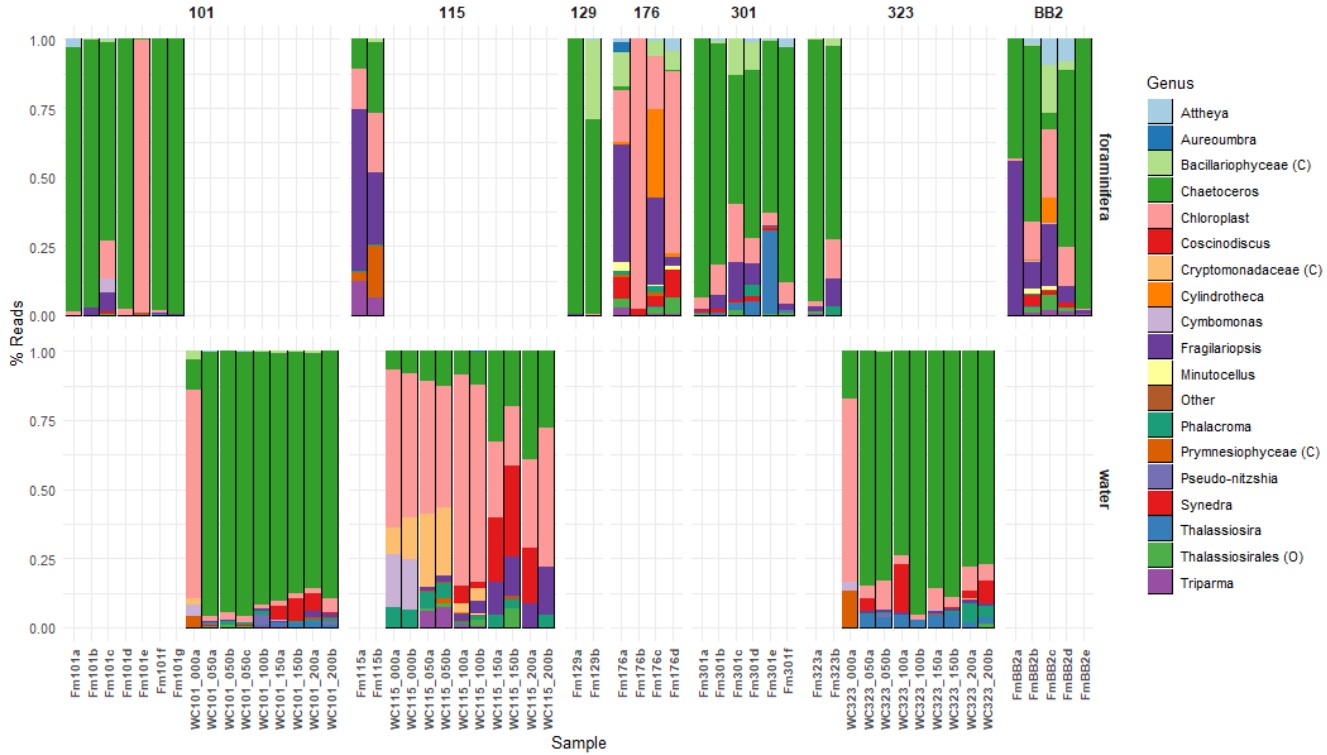

Figure 6. The relative abundance of the 13 chloroplast genera identified with a relative abundance above 2%, along with three classes, and one order identified only to these taxonomic ranks. ASVs with a relative abundance across all samples of less than 2% were grouped in the "Other" category, and ASVs that could not be taxonomically assigned beyond "Chloroplast" are grouped as such. Bars represent individual samples (foraminiferal specimens =Fm and the water column =WC. Numbers above indicate stations separated by columns and provenance is separated by row.

ASV956 was the most abundant chloroplast ASV with a mean relative abundance across all samples of 57.6% when analysing chloroplast ASVs only. This was identified to the level of Class Bacillariophyceae by the SILVA database, and further to the diatom species *Chaetoceros gelidus* (accession NC_063631.1) via a BLAST search of GenBank (100 % identity and coverage) (Fig.6) . ASV956 had a significant Log2 fold change of -2.348 (padj = 0.003443). This was not as significant as the other two chloroplast ASVs (Table A3), probably due to its higher relative abundance in the water column (Fig. 6). ASV956 was common within Baffin Bay with an average relative abundance of 57.2 % (when analysing chloroplasts only) in the water column samples (Fig. 6). ASV956 was therefore relatively common in both the water column and the foraminifera and was clearly a major food source for *N. pachyderma* in Baffin Bay during the summer months.

At Station 101 (west Pikialasorsuaq polynya; Fig. 1), five out of the seven *N. pachyderma* specimens contained > 50 % chloroplast ASVs, and only one specimen contained < 20 % chloroplast ASVs (Fm101a; Fig. 3). When analysing only

chloroplast ASVs, six of the seven foraminifera specimens contained >70 % ASV956 (*Chaetoceros gelidus*). Specimen Fm101e, however, contained over 94 % chloroplast ASVs (472, 548) belonging to two uncharacterised chloroplasts. (Fig. 6). Except for specimen Fm101e, the diatom ASVs at the cooler Station 101 can be said to mirror the diatom population profile in the sub-surface water column where > 85 % of water column chloroplast ASVs were ASV956 (*Chaetoceros gelidus*) (Fig. 5).

The two foraminifera processed at Station 115 (where Atlantic-derived warmer water is found) contained <35 % and <13 % chloroplast ASVs (Fig. 3). Again, this lower chloroplast relative abundance reflected the water column where we found the lowest relative proportion of chloroplast ASVs across the three stations (Fig. 3). *Fragilariopsis* sp. contributed the greatest proportion of chloroplast ASVs (ASV355) in Station 115 specimens (Fig. 6). The proportion of *Fragilariopsis* ASVs were higher in the water column at this station relative to other stations, although ASV956 (*Chaetoceros gelidus*) remained the major diatom ASV present. However, the highest proportion of ASVs were not identified beyond "Chloroplast" at this station. In addition, a non-diatom chloroplast (*Triparma laevis* ASV471) was also present in both the water column and the foraminiferal specimens. This is a relative of the diatoms, and, like most diatom species, forms external siliceous plates. *Triparma laevis* ASVs were only detected in the upper water column at 50-100 m.

Finally, 48 % and 28 % of the ASVs in the two foraminifera from Station 323 were chloroplast ASVs (Fig. 3), and of those, 94.5 % and 69.8 % were ASV956 (*Chaetoceros gelidus*; Fig. 6). This reflected the high proportion of ASV956 in the water column at this station (>73 % across sub-surface samples).

Of the other foraminiferal specimens taken from stations with no comparative water column data, the foraminifera from Station 176 showed the greatest diversity in chloroplast ASVs. Except for Fm176b which had only 3 chloroplast ASVs, Fm176a, c, and d contained a much higher relative proportion of *Fragilariopsis, Synedra* and *Cylindrotheca* diatoms and other chloroplast ASVs. This may indicate higher comparative diatom and algal diversity at this more southerly station.

### 3.3.3 The *N. pachyderma* core microbiome

The core microbiome of *N. pachyderma* is defined here as ASVs found in 80 % of the foraminiferal specimens across all stations. 16S metabarcoding indicated that there were eight core ASVs. Two represented by diatoms: ASV956, *Chaetoceros gelidus* (27/28 specimens) and ASV355, identified in BLASTn as *Fragilariopsis cylindricus* (100 % match to accession NC_045244.1, 24/28 specimens). Then six bacterial ASVs, two from the Flavobacteriaceae family (ASV1392, 24/28 and 1447, 23/28), the genus *Pseudoalteromonas* (ASV 1459, 25/28), the genus *Paraglaciecola* (ASV 122, 25/28), the family Halieaceae (ASV308, 26/28), and genus *Bradyrhizobium* (ASV833, 23/28; Fig. 7). Of these eight ASVs, only the two diatom ASVs (355 and 956) were also significantly more abundant in the foraminifera than the water column, driving the significant differences between the Provenances. This foraminiferal core microbiome made up, on average 47.7 % of the ASVs in the *N. pachyderma* of Baffin Bay, whereas it made up only 9.42 % of ASVs in the water column. Details on the core microbiome including relative abundances and ASV frequencies in the two Provenances can be seen in Table A3.

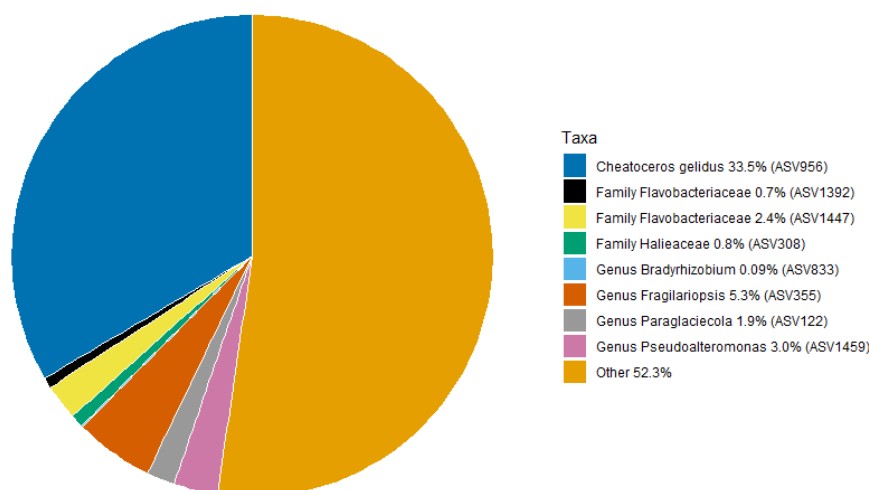

Figure 7. The average relative abundances of the 16S rDNA ASVs found across *N. pachyderma* Type I specimens in Baffin Bay during summer 2017. The average relative abundance of the eight core ASVs (found in ≥ 80 % of specimens) are taxonomically labelled and designated the "core microbiome". These make up 47.7 % of the ASVs in the microbiome. 52.3 % are ASVs found in fewer than 80 % of specimens, designated as non-core and labelled "Other" above. Colour key starts at 12 O'clock and runs anticlockwise.

## 3.4 TEM analysis

TEM imaging was carried out on samples collected during the 2018 cruise (Fig. 1; Table 1) to further investigate the diet/endobionts in this genotype. Whole diatoms, including frustules, were observed within the foraminiferal cell (Fig. 8a). Empty frustules were also observed both inside and outside the foraminiferal cell. Those outside may have been ejected after the diatom organic material was digested/removed (Fig. 8b) or were part of the diatom-derived POM feeding cyst and were likely caught in the external cytoplasm and rhizopodial network at the time of sampling and fixation, as has been reported previously (Spindler et al., 1984).

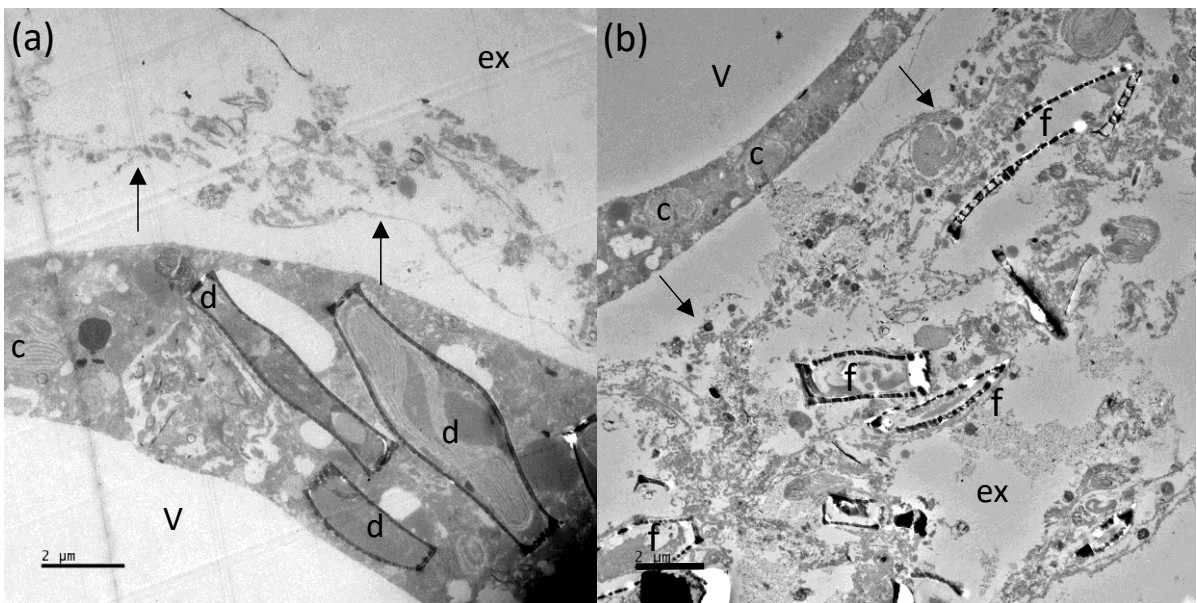

Figure 8. TEM images of *N. pachyderma* (specimen BB2, Table 1) showing internal and external cell and diatom structures. Image **(a)**: Intact diatoms (d) observed inside the specimen cell. Black arrows indicate the cross section of pore plugs in the inner organic lining, v = internal cell vacuole, ex = external to the foraminiferal cell. Image **(b)**: External (ex) to specimen where debris, including empty diatom frustules (f), is apparent. The organic lining is identified by black arrows, c = chloroplasts inside BB2, v = internal vacuole. Large vacuoles were observed in several specimens which may be a result of the fixation process. Scale bars at bottom left are 2µm.

TEM images also show that *N. pachyderma* contains unexpectedly high numbers of chloroplasts throughout the cell from the cell periphery to the cell centre (Fig. 9a-b, Fig. A7). The level of preservation does not allow us to observe the number of membranes surrounding the chloroplasts or the pyrenoid-dissecting lamellae. Nevertheless, although we cannot unequivocally determine the degradation state of all the chloroplasts present, lenticular pyrenoids, and horseshoe-shaped arrays of thylakoid membranes are visible in many chloroplasts (Fig. 9a), as found in *Chaetoceros* spp. (Bedoshvili et al., 2009). There may also be abundant lipid droplets located amongst, and immediately adjacent to the chloroplasts at the cell periphery in some specimens potentially indicating lipid production by the chloroplasts (Jauffrais et al., 2019a, Fig. 9a, Fig. A7a and g).

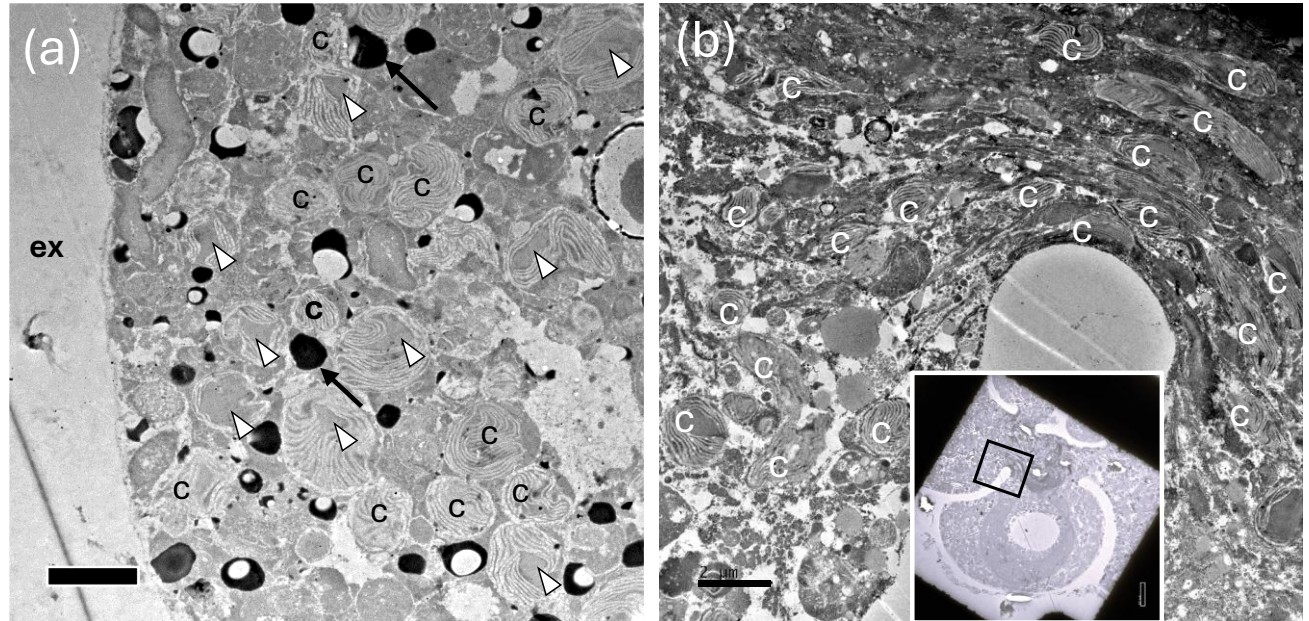

Figure 9. TEM images showing abundant chloroplasts throughout the *N. pachyderma* cell. **(a)** Clusters of chloroplasts (c) within the specimen BB1 cell (Table 1) close to the cell periphery. Those with obvious pyrenoids are marked with white arrowheads. Due to our staining protocol black spots could be lipid droplets or electron opaque bodies (leKieffre et al., 2018;black arrows highlight examples). ex = external to the foraminiferal cell. Scale bar 2µm. **(Inset)** Overview of thin section of specimen BB11 (Table 1) showing chambers and a black square identifying the region of the cell shown in **(b)**. **(b)** Chloroplasts clustered in numbers at the centre of specimen BB11 also appear stretched where the chambers coalesce. Scale bar 2µm.

## 4 Discussion

In this study, our aim was to investigate the microbiome within the polar planktonic foraminifera *N. pachyderma* Type I from the Arctic Baffin Bay region. We defined the microbiome as the combined taxa identified by taxonomic assignment of 16S ASVs generated by metabarcoding. This included food, any endo(sym)bionts, and chloroplast-containing eukaryotes identified by their chloroplast 16S ASVs. Shedding light on their feeding preferences as well as any microbial associations that form part of the "interactome" in the context of the changing climate may afford some clues as to the ability of *N. pachyderma* to withstand/adapt to its rapidly changing environment, and its contribution to the carbonate cycle and ocean alkalinity. For example, eco-physiological and trait-based models indicate that symbiont-barren foraminifera, which *N. pachyderma* is understood to be, are predicted to experience reduced numbers and habitat decline (Roy et al., 2015), and the non-spinose species biomass is likely to be reduced by up to 11 % by 2050 (Grigoratou et al., 2022). Sound knowledge of the eco-physiology

and traits of foraminifera is required for model accuracy, and to that end the genotype and microbiome of the Arctic polar *N.*
*pachyderma* was investigated.

## 4.1 The influence of Station and Depth on the microbial assemblages

Depth was the dominant factor structuring microbial communities in the water column, likely driven by steep vertical gradients in temperature, salinity, and chlorophyll concentration (Fig. 2). This aligns with previous studies reporting depth-stratified microbial assemblages in marine systems (e.g., Zorz et al., 2019; Reji et al., 2020). Because depth data are unavailable for the
foraminiferal samples, its influence on their microbiomes could not be assessed.

In contrast, Station exerted a weaker influence on community composition. Although a significant Station effect was detected in water samples at 50 m depth, the absence of significant pairwise differences and limited replication reduce confidence in this result. Similarly, analyses of the foraminifera-only dataset from the same three stations (F3), and the combined water–foraminifera dataset (FW), revealed minimal spatial structuring, indicating that horizontal variation across these sites was
relatively minor.

The distance–decay relationship, which describes how community similarity declines with increasing geographic distance, is a well-documented pattern in microbial biogeography (e.g., Li et al., 2018). To investigate whether this pattern applied to foraminiferal microbiomes, we expanded our analysis to the full foraminiferal dataset, which spans a broader geographic range. This broader scope revealed significant Station-level differences in microbiome composition. Although only four of 21
pairwise comparisons were significant in the full dataset, these same four comparisons remained significant in a more robustly replicated subset (four stations with ≥4 samples each), where five of six comparisons showed significant differences. This consistency suggests that observed differences at certain stations are robust rather than artefacts of low replication.

Notably, Station 301 consistently differed from Stations BB2, 176, and 101, with 176 and BB2 being geographically distant. Located in Lancaster Sound, Station 301 is influenced by an eastward current carrying Beaufort Sea water (Sanderson &
LeBlond, 1984), which may shape its distinct microbial assemblages prior to mixing with Atlantic-derived waters in Baffin Bay. Interestingly, Station 301 did not differ significantly from its nearest neighbour Station 323, situated at the mouth of Lancaster Sound. These findings suggest that the stronger Station signal observed in the broader foraminiferal dataset is likely attributable to the greater geographic coverage (Fig. 1), consistent with the distance–decay relationship, where community similarity declines with increasing spatial separation.

In addition, selective feeding by *N. pachyderma* may contribute to the observed patterns. Foraminifera retain a core microbiome that accounts for nearly 48 % of their ASVs. Although individuals from Station 301 exhibited significantly different overall microbiome compositions compared to several other stations, their core microbiome did not significantly diverge from those collected elsewhere (Table A3), suggesting that host-level filtering may buffer against environmental variability and reduce the influence of Station. This supports the idea that host selection stabilises key microbial associations
despite broader shifts in community composition. The limited number of significant pairwise differences across the full dataset (four out of 21) may also reflect this buffering effect, alongside subtle or inconsistent geographic influences.

## 4.2 Divergent Feeding strategies in the Neogloboquadrinids

Our findings support previous literature stating that *N. pachyderma* feeds on diatoms (e.g. Spindler and Dieckmann, 1986; Schiebel and Hemleben, 2017, Greco et al., 2021). This study and that of Greco et al. (2021) indicate that *N. pachyderma* feeds predominantly on diatoms (Class Bacillariophyceae), and occasionally other algae (e.g., *Triparmia laevis* - this study). Our TEM micrograph observations of intact diatoms (Fig. 8a) also indicate that *N. pachyderma* feeds on live diatoms and not only detrital (dead) diatoms, as reported by Greco et al. (2021). Our 16S metabarcoding data further suggests that Genus *Chaetoceros* is an important food source for *N. pachyderma* in Baffin Bay. This agrees with Meilland et al. (2024) who cultured *N. pachyderma* Type I from several locations including Baffin Bay by growing them with a diatom food source. They found that specimens grew faster when fed on the genera *Chaetoceros* or *Pseudonitzschia*, compared to *Phaeodactylum* or *Thalassiosira*. However, in their cultures, juveniles provided with *Pseudonitzschia* had the longest life span. Despite this, whilst we recorded *Pseudonitzschia* in the water column and in our foraminiferal specimens (Fig. 6), *Chaetoceros* was by far the dominant genus identified in our specimens from the natural environment.

Our 16S study further revealed that *N. pachyderma* also consumed bacteria, and their 16S ASV composition was significantly different from the water column profile (Table A1). This is most likely driven by the feeding behaviour, where particulate organic matter (POM) is gathered around the test to form a feeding cyst. Once formed, the foraminifer remains within the POM microhabitat, becoming isolated from the water column. This behaviour has already been observed in the Neogloboquadrinids *N. dutertrei* and *N. incompta* (Bird et al., 2018; Fehrenbacher et al., 2018), and in *Globigerinita glutinata* (Spindler et al., 1984).

Although the Neogloboquadrinids all feed within the POM microhabitat, we suggest that they are feeding on very different components within the cyst. Work carried out by Bird et al. (2018) demonstrated that of the three Neogloboquadrinids, *N. incompta* contained the highest proportion of bacterial ASVs (>99.8 %), indicating that it targets the bacteria within the POM microhabitat. The small proportion of chloroplast ASVs (<0.2 %) in the *N. incompta* study indicated that POM was not being passively phagocytosed, but that the bacteria, rather than algae were being specifically selected as food. In contrast, *N. dutertrei* contained only 2-4 % bacterial ASVs and instead maintained a pelagophyte algal endosymbiont population and selectively fed on other protists within the POM. The small proportion of intracellular bacteria in *N. dutertrei* indicated that it too did not specifically phagocytose POM itself. However, in the case of *N. pachyderma*, we found a higher proportion of bacterial ASVs than was identified in *N. dutertrei* and a higher proportion of chloroplast ASVs than found in *N. incompta*, suggesting that *N. pachyderma* may feed on the POM directly for food. This was suggested by Greco et al. (2021), who also demonstrated that the *N. pachyderma* 18S ASV assemblage revealed little difference in intracellular diatom ASVs between the surface dwelling and the deeper dwelling specimens living in diatom-free waters. This finding led them to suggest that *N. pachyderma* feeds on dead diatoms contributing to the sinking detritus, which is supportive of a POM-cyst mode of feeding. However, the TEM images (Fig. 8) in our study indicated that they also feed on living diatoms, as intact diatoms were observed in their cells.

Further, our evidence indicated that a significant component of the *N. pachyderma* diet is active or passive consumption of the bacteria living in the diatom "phycosphere" and diatom-derived POM (Bell & Mitchell, 1972; see Section 4.3).

### 4.3 *N. pachyderma* bacterial ASVs

### 4.3.1 The core microbiome

The core microbiome (defined as ASVs present across 80 % of the foraminifera) could be made up of organisms which (i) the foraminifera specifically target for food, and/or (ii) are routinely passively ingested due to close association with specific food sources, and/or (iii) are endo(sym)bionts. However, the bacteria identified in the *N. pachyderma* core microbiome point to a diatom source and therefore are highly likely to be passively ingested. The core microbiome was composed of just 8 ASVs which accounted for, on average, 47.7 % of the total microbiome. Six bacterial ASVs each contributed small percentages between 0.09 %- 2.96 %, averaging a total of just 8.93% of the core microbiome between them. The bacterial ASV with the highest average relative abundance in the core microbiome (3%) was *Pseudoalteromonas* (ASV1459, Fig. 7, Table A3). *Pseudoalteromonas* are known hydrocarbon degraders (e.g. Calderon et al., 2018). They are the first bacteria to colonise degrading diatom aggregates (Arandia-Gorostidi et al., 2022; Costanzo et al., 2023), and they are known to be algicidal releasing diffusible factors (Costanzo et al., 2023) in the diatom phycosphere. There were also two ASVs from the Flavobacteriaceae family (ASV1447 and ASV1392). This is a large family of bacteria that are widely distributed in the marine environment and are often found associated with detritus (as well as algae, fish and invertebrates; Gavriilidou et al., 2020). Tisserand et al. (2020) isolated ten species from Baffin Bay, and all were shown to grow on exudates (dissolved organic matter) from two Arctic diatoms (*Fragilariopsis cylindricus* and *Chaetoceros neogracilis*). Since members of the Flavobacteriaceae and *Pseudoalteromonas* are shown to co-occur with diatoms (Amin et al, 2012) it is likely that *N. pachyderma* passively consumes these bacteria as it feeds on diatom detritus (Greco et al., 2021) and on living diatoms. A BLASTn search (NCBI) identified ASV1392 as 99.6 % identical to a Flavobacteriaceae of the genus *Tenacibaculum* including *T. insulae, T. haliotis and Tenacibaculum* sp. This genus contains many opportunistic fish pathogens, some of which are found to target fish teeth, a high source of calcium shown to promote the bacteria's growth (Hikida et al., 1979; Frisch et al., 2018). Growth promotion by calcium may be a common feature of the *Tenacibaculum* genus and may be another reason why this ASV is identified with *N. pachyderma,* and the calcite tests of foraminifera may provide a suitable niche for this genus. Another core ASV was 308, attributed to the OM(NOR) genus of the Family Halieaceae, order Cellvibrionales (Spring et al., 2015). The order Cellvibrionales are gram-positive aerobes that are mesophilic and neutrophilic chemoorganotrophs. However, some members of the Family Halieaceae may additionally be capable of aerobic photoheterotrophic growth using bacteriochlorophyll a, and carotenoids for the harvesting of light. Several strains may also be able to use proteorhodopsin to utilise light as an energy source (Spring et al., 2015). *Paraglacieocola* (ASV122) was another core ASV (Fig. 7). Numbers of ASVs and the relative abundances were substantial and similar between the Provenances (Table A3). *Paraglacieocola* are a genus of the family Alteromonadaceae. In a BLASTn search this ASV shows 100 % identity with *Paraglaciecola psychrophila, P. arctica* and

several other *Paraglaciecola* sp. sequences. *Paraglaciecola psychrophila* is a gram-negative, psychrophilic, motile rod-shaped bacteria. Identified from the sea ice of the Canadian Basin and the Greenland Sea, it is aerobic, and optimum growth is at 12°C (Zhang et al., 2006). Unable to reduce nitrate, it may be associated with POM as an N-source and so be ingested by *N. pachyderma* as it feeds on the detritus. The final core ASV *Bradyrhizobium* ASV833 constituted on average only 0.09 % of

foraminiferal and 0.004 % of water column ASVs. *Bradyrhizobium* contains mainly nitrogen fixing species that are part of phylogenetic subcluster IK of *nifH* (Chien and Zinder, 1994; Gaby and Buckley, 2014; Fernández-Méndez et al., 2016), which encodes the nitrogen fixing enzyme nitrogenase. Sequences from subcluster IK can make up >50 % of the *nifH* sequence abundance in the open waters of the Central Arctic Ocean (Fernández-Méndez et al., 2016), supporting our identification of this ASV in the polar waters of Baffin Bay.

**4.3.2 The differentially abundant ASVs and PICRUSt2 pathways**

Metabolic pathway abundances were predicted using PICRUST2. It is important to note that these are predictions indicating potential functional capacity and are not indicative of active processes. Nevertheless, many of the predicted pathways are consistent with the hypothesis that the bacteria in the foraminiferal microbiome are derived from foraminiferal feeding on diatoms, and diatom-derived POM.

A POM feeding cyst will contain an oxygen gradient with oxygen concentrations in the centre significantly below ambient seawater (Alldredge and Cohen 1987). Interestingly, facultatively anaerobic fermentation pathways are present in higher abundance in the foraminiferal microbiome, which may be a result of preferential ingestion of fermenting bacteria due to their presence in the centre of a low oxygen POM feeding cyst. There are two pathways categorised as "Other biosynthesis" pathways (P125-PWY and PWY-7391, Fig. 5) that are involved in the biosynthesis of the antifreeze butanediol, via the

fermentation of pyruvate (Caspi et al., 2014). These pathways are attributed to 49 ASVs predominantly from Class Gamma- and Alphaproteobacteria, Phylum Firmicutes, and Phylum Actinobacteriota, and include two of the differentially abundant ASVs in the foraminiferal microbiome (Fig. 4; Table A2). Four other identified fermentation pathways utilise monosaccharides to produce ATP and reducing power (NADH). These monosaccharides are abundant in the diatom exopolysaccharide (EPS) exudates in the phycosphere (Daly et al., 2023) and therefore, again, the phycosphere or diatom derived POM is likely to be

the foraminiferal microbiome source of these fermenting bacteria. Four hundred and ninety-six ASVs contribute to this pathway, with three of the differentially abundant ASVs doing so (Table A2).

Peptidoglycan synthesis pathways are also more abundant in the foraminiferal microbiome, this is driven by 897 ASVs, of which four are also significantly differentially abundant in the foraminiferal microbiome (Table A2). A pathway for synthesis of a single amino acid L-lysine (PWY-2941) is more prevalent here, compared to higher abundance of three different amino

acid biosynthesis pathways in the water column assemblage. L-lysine is essential for cell wall biosynthesis (Gillner et al., 2013) and this pathway also produces meso-diaminopimelate which is another component of the peptidoglycan cell wall (Weinberger and Gilmar 1970). Supporting this, there are four further peptidoglycan synthesis and recycling pathways (grouped in the category "Cell structure biosynthesis", Fig. 5). Pathway PWY-6471 is specific to gram-positive bacteria, and

the greater abundance of these cell wall synthesis pathways in general might indicate a greater relative abundance of gram-positive bacteria in the foraminiferal microbiome compared to the water column, although at present it is unclear why.

The degradation of certain carbohydrates was also key in the foraminiferal microbiome with 624 ASVs associated with these pathways. For example, pathways for the degradation of the polysaccharides glycogen and starch, and the sugars sucrose, glucose and xylose were differentially abundant, and given these are abundant sugars in the diatom phycosphere (Daly et al., 2023), this is further supporting evidence that the majority of bacteria in the foraminiferal microbiome are derived from the phycosphere or diatom-derived POM.

Of interest were two additional pathways. The first was norspermidine biosynthesis (PWY-6562) which plays a central role in biofilm formation in *Vibrio spp*. (Wotanis et al., 2017), whilst inhibiting biofilm formation by other species, and in particular other gram-negative bacteria (Qu et al., 2016). Norspermidine biosynthesis was identified across seven Gammaproteobacterial ASVs, including three Alteromonadales and four Vibrionales, one of which (ASV116, genus *Vibrio*) was differentially abundant in the foraminiferal microbiome.

Finally, interestingly the palmitate biosynthesis II pathway (PWY-5971) is more abundant in foraminiferal microbiome. Palmitate is a long-chain saturated fatty acid. It is produced both by algae such as diatoms and by bacteria (Allan et al., 2023), but in our data set the ASVs responsible for this pathway are one ASV of the Genus *Alteromonas*, and three ASVs from the Genus *Cellvibrio*. Palmitate, or palmitic acid, is a very abundant saturated fatty acid and key precursor for phospholipids and lipopolysaccharides essential for the bacterial plasma membrane (Cronan & Thomas, 2014). Given the universal nature of this requirement, it is surprising that genes encoding enzymes in this pathway are more abundant in the foraminiferal microbiome. Interestingly however, palmitic acid is used by pathogenic bacteria to modify their proteins and glycoproteins to avoid detection by host immune system TLR4 receptors (Toll-Like-Receptor family) (Sobocińska et al., 2018) thereby increasing infectivity and the production of biofilms. However, TLR4 evolved 500 million years ago near the beginning of vertebrate evolution (Beutler & Rehli, 2002) and is not known to be present in protists. Therefore, the reason for increased abundance of the palmitate biosynthesis II pathway in the foraminiferal microbiome compared to the water column is currently unknown.

### 4.4 *N. pachyderma* chloroplast ASVs

Two diatom chloroplasts contributed 5.3 % (ASV355, *Fragilariopsis cylindricus*) and 33.46 % (ASV956 *Chaetoceros gelidus*) of the core microbiome (Fig. 7). Both also contributed to the significant difference in assemblage composition between the foraminifera and the water column (Fig. Table A2). Chloroplasts were exceptionally abundant in our TEM images (Fig. 9 and Fig. A7), which appear very similar to observations made in kleptoplastic benthic species (e.g. Bernhard & Bowser, 1999; Jauffrais et al., 2018; Jesus et al., 2022). This raises important questions about the nature of the relationship between the chloroplasts and *N. pachyderma* Type I and is discussed below. Empty diatom frustules were also observed in the TEM images, which is highly consistent with previous reports that diatoms are a significant part of the *N. pachyderma* diet (e.g. Hemleben et al., 1989; Scheibel & Hemleben, 2017; Greco et al., 2021).

On average 53.3 % of all 16S rDNA ASVs in the foraminifera belong to chloroplast-containing taxa (Fig 2; Sect. 3.3.2). This contrasts with the 3.51 % average proportion found in the water column. Most of the foraminiferal intracellular chloroplast ASVs, are dominated by ASV956, *Chaetoceros gelidus* (BLASTn) from the diatom class Bacillariophyceae. The compositional dominance of ASV956 in the foraminifera reflects the chloroplast ASV composition of the water column (Fig. 6), although found in much higher proportions in the foraminifera (Fig. 3). The presence of ASV956, *Chaetoceros gelidus,* in all other specimens indicates its importance to *N. pachyderma* in this location and season. It is characteristic of northern temperate and polar waters (Chamnansinp et al., 2013), and it is a known important biomass fraction in Baffin Bay (Crawford et al., 2018). In fact, eight strains were isolated from Baffin Bay only during bloom development or bloom peak (Ribeiro et al., 2020) and *Chaetoceros's* reputation for bloom forming (Booth et al., 2002) is reflected here in its high abundances compared to other species.

The diatom chloroplast 16S ASVs identified in this study (Fig. 6) are also consistent with the diatoms found by Greco et al. (2021), who identified *Chaetoceros* and *Fragilariopsis* as major components of the *N. pachyderma* 18S ASVs from Baffin Bay. Both *Chaetoceros* (ASV1413, ASV956) and *Fragilariopsis* 16S ASVs (ASV355) were amongst those ASVs driving the significant difference between the foraminifera and the water column, and both are major constituents of the core microbiome. Intact *Fragilariopsis* were also identified in the foraminiferal TEM images (Fig. 8a) hinting that, like *Ammonia* sp. and the miliolid *Hauerina diversa, N. pachyderma* may phagocytose the entire diatom before digesting the cell and then extruding the silicate frustules (Jauffrais et al., 2018; Pinko et al., 2023).

**4.5 Observation of abundant chloroplasts throughout the cytoplasm of *N. pachyderma* Type I**

To our knowledge this is the first report of large numbers of chloroplasts observed by TEM imaging and recorded via metabarcoding in any planktonic foraminiferal species. These observations cover two summers in different regions of Baffin Bay. The high numbers observed, and the relative abundance of diatom chloroplasts recorded, could indicate a kleptoplastic behaviour in *N. pachyderma* Type I*,* a strategy that is well known in several protist lineages such as benthic foraminifera (e.g. Jesus et al., 2022), dinoflagellates (e.g. Takano et al., 2014; Yamada et al., 2023), and ciliates (e.g. Johnson et al., 2007). Kleptoplasty refers to the phenomenon where an organism sequesters chloroplasts from its microalgal prey. The original definition does not include the requirement for temporary photosynthesis to continue in the host (Clark et al., 1990; Jauffrais et al., 2018) and is appropriate since chloroplasts are known to perform many functions in addition to photosynthesis. These include amino acid, nucleotide, and fatty acid synthesis as well as N and S assimilation (e.g. Cedhagen, 1991; Bobik and Burch-Smith, 2015). Further, the benthic foraminiferal species *Nonionellina labradorica* retains chloroplasts despite living in sediments below the photic zone. The photosynthetic pathway of their retained chloroplasts is therefore not functional (Cedhagen, 1991; Jauffrais et al., 2019b) and the reason for chloroplast retention in this, and other species that live in the aphotic zone, is unknown (Bernhard and Bowser 1999). However, its importance is reflected by the discovery that the kleptoplast genome in *Nonionella stella*, another benthic species that lives below the photic zone, is transcribed in the host

(Gomaa et al., 2021; Powers et al., 2022). Where kleptoplasts do continue to photosynthesize in the new host, evidence suggests that this has been important in supporting major evolutionary innovations crucial to the current ecological roles of such protists in the marine environment (Stoecker et al., 2009). Therefore, the role of the retained chloroplasts remains a fascinating question in many species of foraminifera, including *N. pachyderma* Type I, and it is important to assess and further investigate potential kleptoplast roles to understand any contribution they make to *N. pachyderma* evolution, successful ability to inhabit the true polar habitat, and evaluate its potential resilience to future climate change.

### 4.5.1 Evidence for potential kleptoplasty in *N. pachyderma* Type I

Foraminifera such as *N. pachyderma* that eat diatoms (and other algae) would be expected to contain some chloroplasts in their cytoplasm as a byproduct of their grazing. For example, 18S metabarcoding demonstrates that the non-kleptoplastic benthic foraminifer *Ammonia* sp. (Jauffrais et al., 2016)*,* grazes on diatoms in a comparable way to the kleptoplastic *Elphidium* sp. and *Haynesina germanica*, (Chronopoulou et al., 2019), and chloroplasts are indeed observed within the cytoplasm of *Ammonia* sp. Yet the relative plastid abundance in *Ammonia* sp., is reported as "rare" compared to "abundant" in *Elphidium* sp. and *Haynesina* sp. (Goldstein et al., 2004; Cesbron et al., 2017; Jauffrais et al., 2018), with a high proportion of chloroplasts in *Ammonia* sp. undergoing degradation (Jauffrais et al., 2018; LeKieffre et al., 2018). In contrast, our TEM images show high numbers of chloroplasts in *N. pachyderma*, congruent with or greater than the abundance observed in the TEM images of the kleptoplastic foraminifera such as *Elphidium* sp. and *H. germanica* (Jauffrais et al., 2018; Fig. 9, Fig. A7).

Kleptoplasty is common amongst benthic foraminifera (e.g. Lopez et al., 1979; Lee et al., 1988; Cedhagen 1991; Tsuchiya et al., 2018; Jauffrais et al., 2018; Pinko et al., 2023). The molecular studies identifying the source of kleptoplasts in benthic foraminifera to date would suggest a diatom source from the family Thalassiosiraceae, but potentially, kleptoplasts from more than one diatom species can be present (e.g. Pillet et al., 2011; Lechliter 2014; Jauffrais et al., 2019a; Tsuchiya et al., 2020; Pinko et al., 2023). More than 20 diatom species have been identified in benthic foraminifera that host intact diatom symbionts, (Lee 1995; Schmidt et al., 2018), with potentially up to three different symbionts within a single foraminifer at the same time (Lee, 2011). In addition, diatom symbiont shuffling appears to be an adaptation to changing environmental conditions such as heat stress (Schmidt et al., 2018). These studies indicate that host-symbiont or host-kleptoplast relationships are not strictly species-specific, supporting our findings of multiple diatom ASVs.

The chloroplasts in *N. pachyderma* are distributed throughout the foraminiferal cytoplasm (Fig. 9). In the benthic kleptoplastic species, chloroplast location is specific to the foraminiferal host species, where some kleptoplasts may be associated with the cellular periphery, while others, as observed here, may be distributed throughout the cell cytosol (e.g. Jauffrais et al., 2018; Pinko et al., 2023). Chloroplast placement therefore cannot provide a clear-cut indicator of kleptoplasty.

The degradation state of the *N. pachyderma* chloroplasts in this study is uncertain due to poor fixation of the samples, and therefore, unfortunately cannot provide information on how intact they are. However, in benthic foraminifera, actively photosynthesising kleptoplasts can remain active from just a few days to a few months before being digested (e.g. Grzymski

et al., 2002; Jauffrais et al., 2018). Given that turnover rates are extremely variable, the number of degrading versus intact kleptoplasts must also be highly variable from species to species.

It is important to note that no photosynthetic potential was found in six *N. pachyderma* Type VII individuals from the North Pacific using fast repetition rate (FRR) Fluorometry (Takagi et al., 2019). There was also no evidence of non-functional chlorophyll using this technique (Takagi et al., 2019). This is extremely surprising given the herbivorous nature of *N. pachyderma* (e.g. Spindler and Dieckmann, 1986; Schiebel and Hemleben, 2017; Greco et al., 2021; this study), and may reflect different feeding strategies in the two genotypes, whereby Type I retains chloroplasts but Type VII does not, or Type VII has a broader ranging diet and so there is no resultant build-up of chloroplasts in the cytoplasm. *Neogloboquadrina pachyderma* Type I should now be tested using FRR fluorometry to identify whether retained chloroplasts have photosynthetic potential and therefore behave as traditional kleptoplasts or not. This potential difference could represent a divergent evolutionary adaptation in *N. pachyderma* Type I to survive and flourish in the extreme Arctic environment. *Neogloboquadrina pachyderma* has genetically diversified to inhabit a wide range of extreme environments from the Arctic and Antarctic polar waters to the frontal and upwelling systems of the transitional to tropical zones (e.g. Darling et al., 2008). Type I *N. pachyderma* diverged from its Southern Ocean counterparts during the early Quaternary (Darling et al., 2004), allowing substantial time for distinct adaptations to develop in its North Atlantic and Arctic habitat.

**4.5.2 Potential chloroplast storage to facilitate overwintering and reproduction**

The cytoplasmic chloroplasts observed in *N. pachyderma* Type I (Fig. 9) are retained from their diatom food source, in particular from *Chaetoceros* spp. (Booth et al., 2002; this study). These chloroplasts represent a rich source of amino acids, fatty acids, lipids, vitamin E, pro-vitamin A, lutein, Cu, Fe, Zn and Mn (Gedi et al., 2017). Therefore, either functioning photosynthetic kleptoplasts and/or chloroplasts themselves could potentially provide *N. pachyderma* Type I with a substantial additional energy resource in the challenging Arctic environment. If chloroplasts can be retained in the cytoplasm over many months before consumption, they could provide a valuable source of nutrition for the overwintering population. A similar overwintering survival strategy citing the high nutrient levels of stored chloroplasts has been proposed for the benthic foraminifera *N. labradorica* (Salonen et al., 2021), as it is known that no photosynthesis occurs in these kleptoplasts in the winter months (Ceghagen 1991).

Ecological processes in the Arctic are largely governed by sea ice and light dynamics. There is a general perception of minimal biological activity in the Arctic marine surface layers during the Arctic winter, due to the low light intensity producing minimal photosynthetic activity. However, studies around Svalbard in January 2012-2015 revealed unexpectedly high biological activity in the Arctic winter, with high respiration rates per unit of biomass in the upper 100 m water column (Berge et al., 2015a, 2015b; Falk-Petersen et al., 2015), and an earlier winter *Calanus* copepod (Arthropoda) presence than previously thought (Espinasse et al., 2022). In Baffin Bay, low but significant phytoplankton growth was also observed during winter under the sea ice at extremely low light levels (Randelhoff et al., 2020). Since *N. pachyderma* Type I are thought to feed on both POM (including Arthropoda) and live diatoms (Greco et al., 2021 and this study), such wintertime POM-producing

biological activity combined with stored chloroplasts (whether photosynthesising or not) could provide significant nutritional resources for an overwintering population of foraminifera.

These factors potentially combine to provide *N. pachyderma* with a significant nutritional resource to survive over the winter months, but questions remain about its behaviour in the water column and the form in which it may overwinter. Sediment traps in the Irminger Sea indicate a very low-level population of overwintering *N. pachyderma* and their isotopic signature profiles imply that a dormant noncalcifying population of *N. pachyderma* may remain in the water column during winter (Jonkers et al., 2010). However, it is possible that the *N. pachyderma* population they detected may not fully represent the true winter

population size, since sieve sizes of 150 µm would not retain smaller mature/immature *N. pachyderma* specimens. Potentially, *N. pachyderma* could also remain buoyant in the water column as non-reproducing immature cells, slowing down their cellular metabolism as largely quiescent cells during the most challenging winter months. In culture, several specimens of *N. pachyderma* Type I exhibited extended periods of dormancy or inactivity, followed by recovery (Westgård et al, 2023).

### 4.6 Palaeoenvironments and geochemical signatures

The biological adaptations and interactions of calcifying foraminifera have varying influences on the geochemistry of their test, as photosymbionts are known to influence test geochemistry (e.g. Spero et al., 1991; Bemis et al., 1998; 2002; Anand et al, 2003; Russell et al., 2004), and symbiont-host respiration and potentially respiration of endobiont bacteria may increase the use of metabolic (respired) carbon in their tests (e.g. Rink et al., 1998; Wolf-Gladrow et al., 1999; Hönisch et al., 2003; Eggins et al., 2004; Bird et al., 2017). To fully understand variations in the geochemical signatures of Arctic *N. pachyderma* tests

through time in the fossil record, we need to improve our understanding of the ecology and interactions between *N. pachyderma* and the intracellular microorganisms which it hosts. Interactions may exhibit ontogenetic or strong seasonal differences and may be facultative or obligate. Recent geochemical studies have used high-resolution single-specimen and even single chamber analyses to investigate both the biological and seasonal influences on test geochemistry throughout the lifetime of calcareous foraminifera (e.g. Spindler and Dieckmann, 1986; Takagi et al., 2015, 2016; Lougheed et al., 2018; Pracht et al., 2019; Metcalfe

et al., 2019). Single test analysis of $\delta^{18}O$ isotope values has identified two distinct populations of morphologically identical *N. pachyderma* populations in the North Atlantic during the last deglacial period. Isotope values indicate a temperature difference of about 4°C, potentially due to a bi-modal seasonal population with peak abundances separated temporally in late spring/early summer and late summer (Brummer et al., 2020). Spatial difference in the assemblage water depth, driven by low salinity meltwater (Brummer et al., 2020) may also contribute towards these seasonal differences. Since potential kleptoplasty (this

study) could occur seasonally, obligately or facultatively, in *N. pachyderma* Type I, it is imperative to understand if the stored chloroplasts photosynthesise or not because photosynthesis is known to influence $\delta^{18}O$ values (e.g. Spero & Lea 1993; Bemis et al., 1998).

# 5 Conclusions

The *N. pachyderma* Type I microbiome consisted of a range of bacterial ASVs that are significantly different from those of
the water column. The genera profile and the putative metabolic pathways identified imply that the likely source of the bacterial
ASVs in the foraminiferal microbiome derive from the phycosphere of their diatom food source, or from the diatom-derived
POM. Although *N. pachyderma* Type I clearly utilises diatom-associated bacteria as a food source, it is most likely passively
consumed during ingestion and digestion of the diatom prey and diatom-derived POM. *Neogloboquadrina pachyderma* Type
I also retains large numbers of intact diatom chloroplasts in its cytoplasm. Whilst our TEM images cannot account for the
degradation state of the chloroplasts, they appear commensurate with TEM images derived from kleptoplastic benthic
foraminifera. 16S metabarcoding data suggests that most of the chloroplasts inside *N. pachyderma* Type I across Baffin Bay
during the summer of 2017 derived from the diatom *Chaetoceros gelidus*, and that these comprised the majority component
of the core microbiome at that time. Work still remains to understand the relationship between the chloroplasts and the
foraminiferal "host" to determine the chloroplast role in the nutrition of *N. pachyderma* Type I, be it photosynthesis, a
facultative nutrient-rich overwintering store, or simply a build-up of chloroplasts due to gorging on diatoms. Improving our
understanding of the biology, ecology and seasonal microbial interactions of Arctic *N. pachyderma* is essential to disentangle
the palaeoproxies for this species and develop an understanding of its susceptibility/adaptability to climate change in the rapidly
contracting Arctic biome.

## 6 Appendix

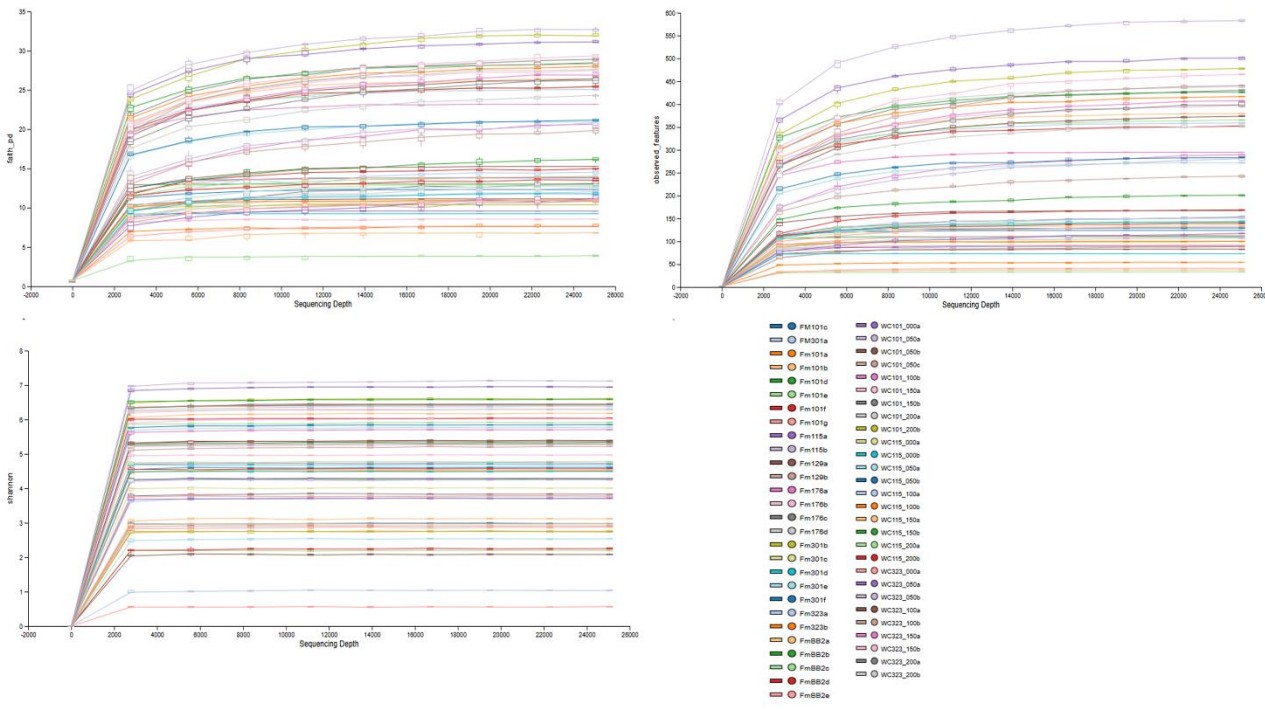


Figure A1. Alpha rarefaction curves of all foraminiferal and water column samples (Faith PD, Observed features and Shannon). The levelling out of the curves demonstrates that the sequencing depth is adequate for all samples.

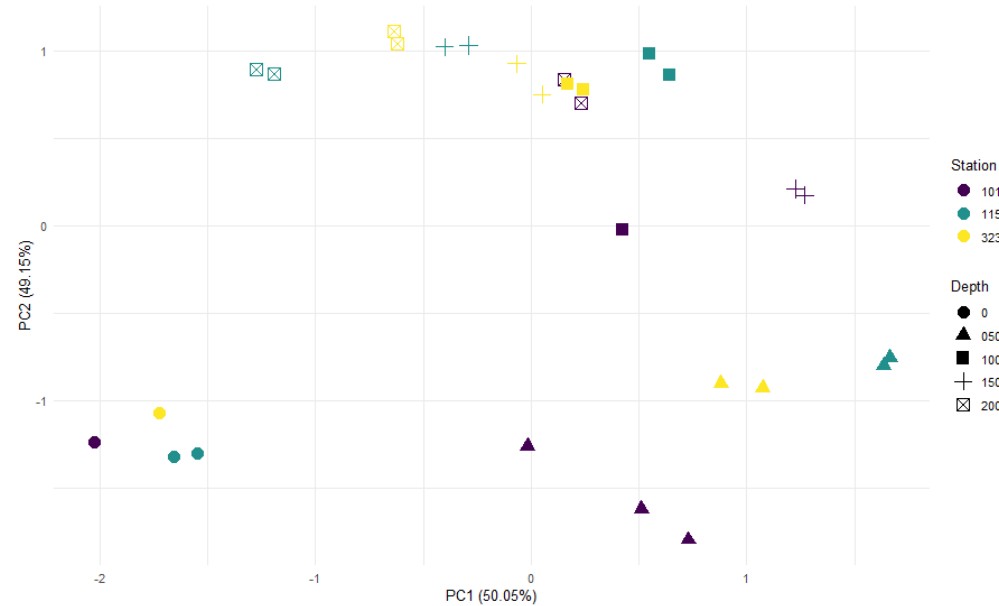


Figure A2. Aitchison dissimilarity PCA plot of water samples from different depths at three stations. Colours represent Station and shapes represent Depth. Depth drives 55 % of the variability (Pr = 0.001) compared to Station driving 6.9% of the

variability (Pr = 0.458, *Adonis2,* Vegan*).*

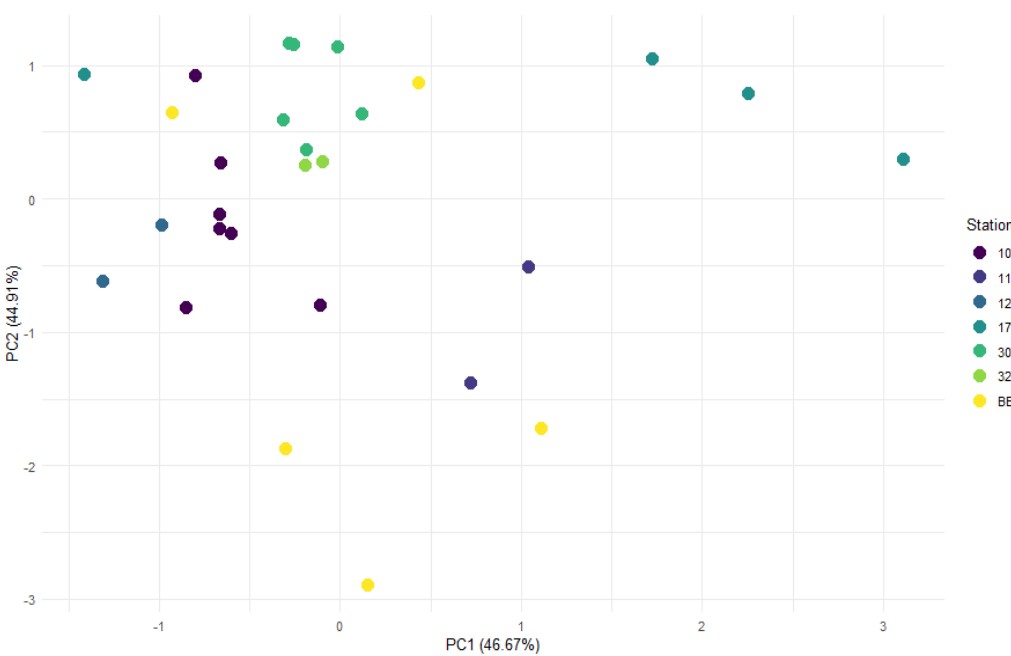


Figure A3. Aitchison dissimilarity PCA ordination of foraminiferal samples across all stations. There is a degree of clustering of foraminifera by station with 48.3% of the variation in the foraminifera driven by Station (Pr = 0.003; *Adonis2,* Vegan).



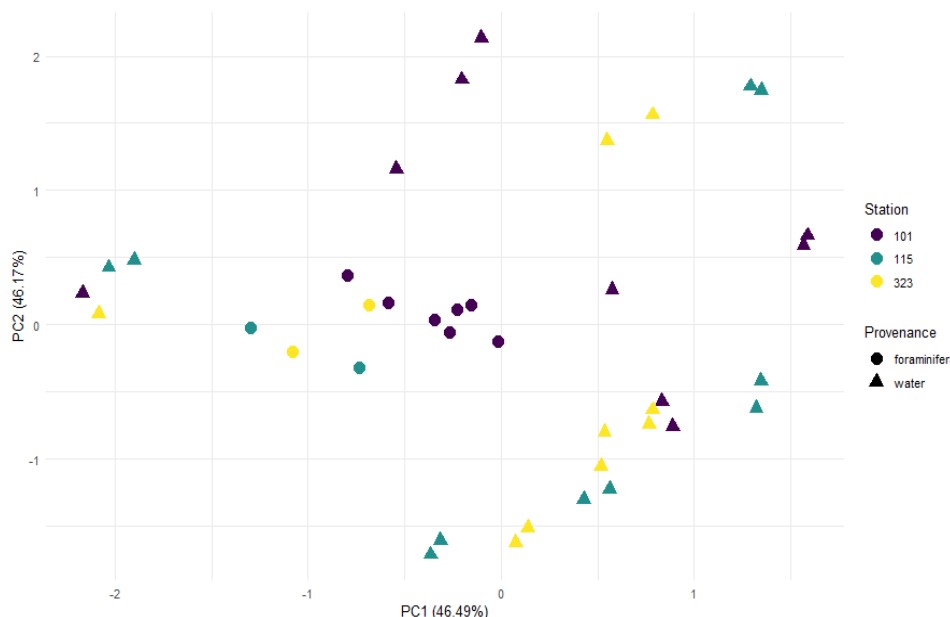

Figure A4. Aitchison dissimilarity PCA ordination of the foraminiferal and water column samples (dataset FW). Provenance
drives 9.7% of the variation and the microbial assemblage between Provenances is statistically different (Pr = 0.013; *Adonis2,*
Vegan).


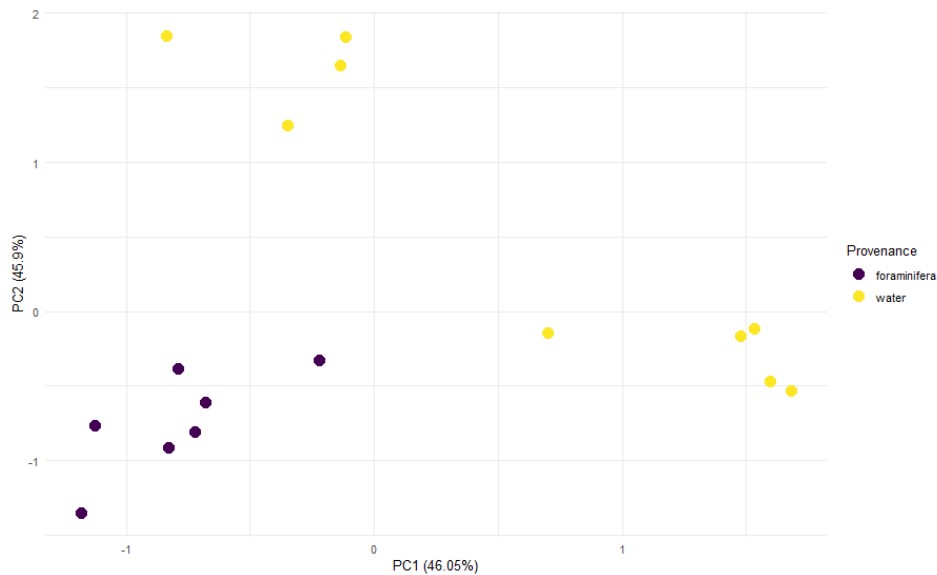

Figure A5 Aitchison dissimilarity PCA plot of water column (yellow) and foraminiferal samples (purple) from Station 101. 41.7 % of the differences in ASV composition are driven by Provenance (Pr = 0.002; *Adonis2,* Vegan).


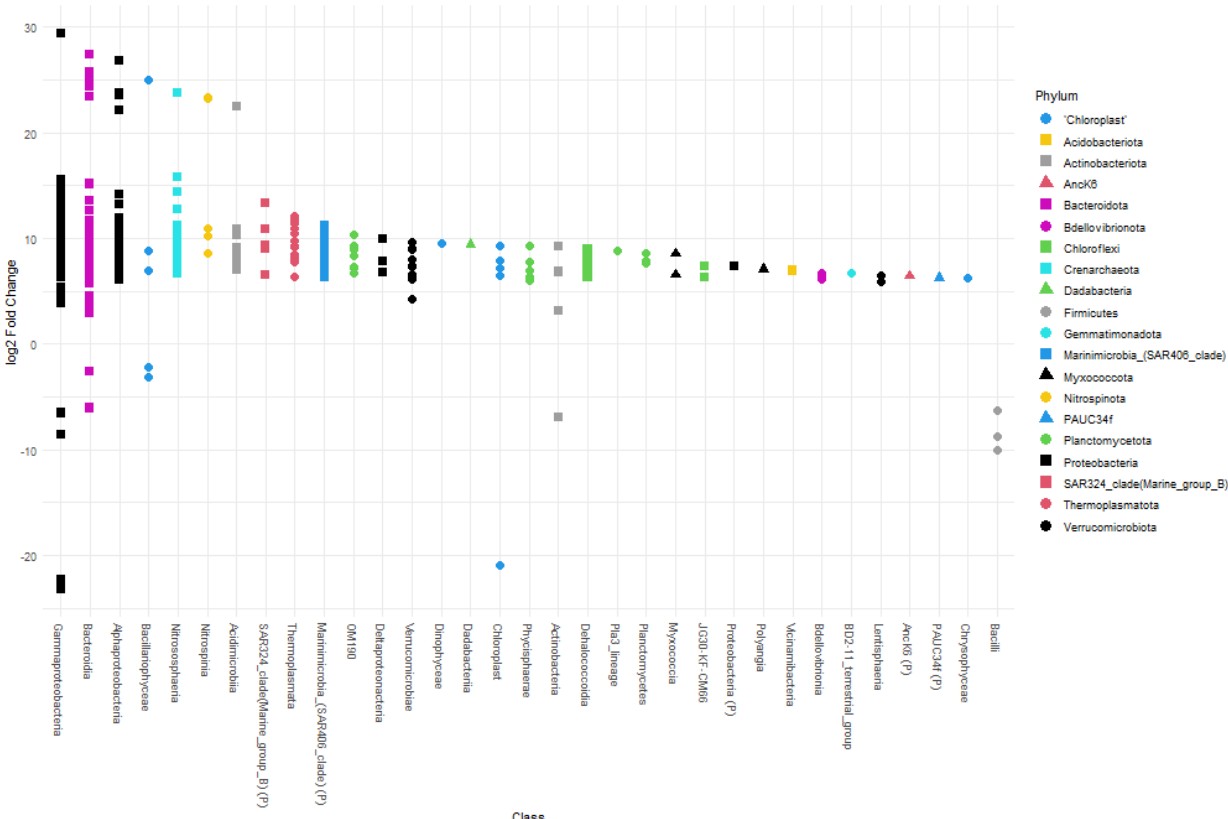


Figure A6. Differential abundance testing of ASVs between Provenances at Station 101 using *DESeq2*. The Log2 fold change in ASVs is the log-ratio of the ASV means in the water column and foraminifera. ASVs with positive Log2 fold change are significantly more abundant in the water column assemblage and ASVs with negative values indicate ASVs that are significantly more abundant in the foraminiferal assemblages. The Class, or the highest level of taxonomic assignment available for each ASV is given on the x-axis. (P) =Phylum



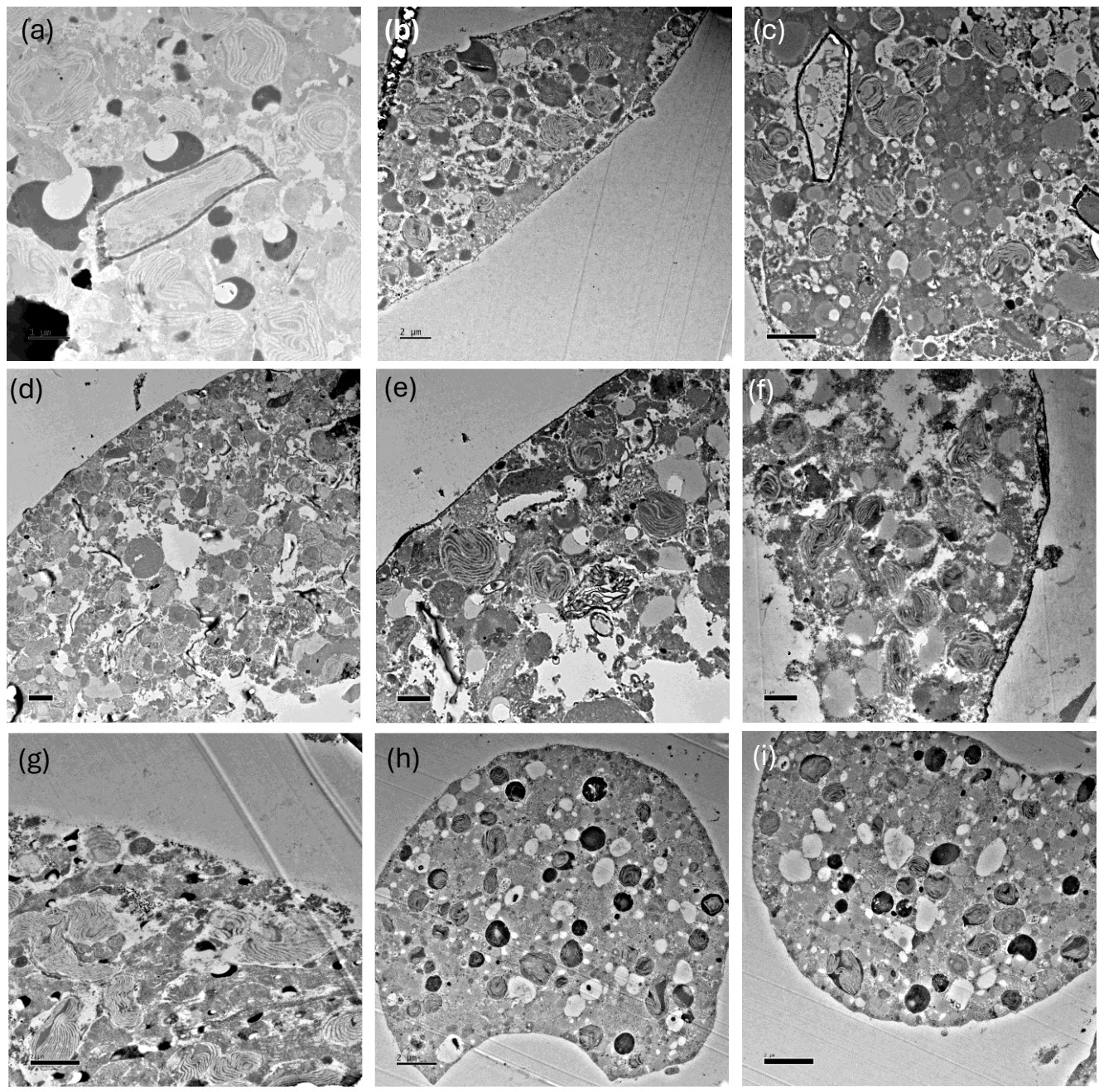


Figure A7. TEM images of *N. pachyderma* specimens (a) BB1 (b) BB9B, (c) BB11, (d)-(f) BB8, (g) BB9C and (h)-(i) BB12. Scale bars are 1 µm ((a), (e), and (f) all others are 2µm. TEM imaging shows that chloroplasts were observed in all fixed specimens.

Table A1. Results of PEMRANOVA (*Adonis2*, Vegan) and tests test for homogeneity of multivariate dispersions on robust Aitchison distance matrices of foraminifera and water column samples. Dataset FW is all foraminifera and all water samples from Stations 101, 115 and 323. Dataset 101 is all water column and foraminiferal samples from Station 101. Dataset F3 is all foraminifera from Stations 101, 115 and 323.

| Test on | Res.Df | Df | Sum of Squares | Mean Sqs | F. Model | R2 | Pr(>F) | Homogeneity of multivariate dispersions significance Pr(>F) |
|---|---|---|---|---|---|---|---|---|
| Water only: Depth | 23 | 4 | 31.72 | 7.93 | 7.17 | 0.55 | 0.001 | 0.068 |
| Water only: Station | 25 | 2 | 3.98 | 1.36 | 0.94 | 0.069 | 0.458 | 0.514 |
| 50m water samples: Station | 4 | 2 | 13.87 | 7.02 | 110.41 | 0.98 | 0.012 | 0.442 |
| Dataset FW: Provenance | 37 | 1 | 8.26 | 8.26 | 3.99 | 0.097 | 0.013 | 0.001 |
| Dataset FW: Station | 36 | 2 | 4.58 | 2.09 | 1.03 | 0.054 | 0.423 | 0.291 |
| Dataset 101: Provenance | 14 | 1 | 14.7 | 14.7 | 10.00 | 0.417 | 0.002 | 0.033 |
| Forams only: Station | 21 | 6 | 30.83 | 5.14 | 3.26 | 0.483 | 0.003 | 0.131 |
| Forams only at 101, 176, 301 and BB2 (robust dataset where n≥4): Station | 18 | 3 | 20.98 | 2.48 | 4.19 | 0.41 | 0.003 | 0.087 |
| Dataset F3 Forams only at 101, 115 and 323: Station | 8 | 2 | 14.10 | 3.84 | 5.22 | 0.566 | 0.007 | 0.039 |

Table A2. The ASVs demonstrating significantly higher differential abundance in the foraminiferal samples compared to the water column samples based on DESeq2 analysis of the FW and 101 datasets (See section 3.2.3). The contribution of the bacterial ASVs to differentially abundant pathways identified in PICRUSt2 (Section 3.2.4) is also shown.

| Phylum | Family | Genus | Species | ASV# | log2 fold change | p-adjust | PICRUSt2 pathways contribution |
|---|---|---|---|---|---|---|---|
| **ASVs differentially abundant in FW dataset only** | | | | | | | |
| Proteobacteria (Gamma) | Saccharospirillaceae | *Oleispira* | NA | 420 | -5.75999 | 4.0E-02 | Carbohydrate degradation |
| Chloroplast | Chloroplast | Chloroplast | Chloroplast | 1538 | -23.36711 | 2.7E-14 | |
| Actinobacteriota | Nocardiaceae | *Gordonia* | NA | 1403 | -5.54588 | 3.5E-02 | Peptidoglycan synthesis, antifreeze production, carbohydrate degradation |
| **ASVs differentially abundant in FW and 101 dataset** | | | | | | | |
| Firmicutes | Streptococcaceae | *Streptococcus* | NA | 1402 | -24.62896 | 1.2E-13 | Peptidoglycan synthesis, and carbohydrate degradation |
| Firmicutes | Streptococcaceae | *Streptococcus* | *S. salivarius* | 609 | -9.35070 | 3.6E-14 | Peptidoglycan synthesis, fermentation and antifreeze production |
| Firmicutes | Streptococcaceae | *Lactococcus* | *L. lactis* | 391 | -6.10004 | 4.2E-04 | Peptidoglycan synthesis and carbohydrate degradation |
| Proteobacteria (Gamma) | Moraxellaceae | *Acinetobacter* | NA | 927 | -6.86770 | 2.7E-03 | None identified |
| Proteobacteria (Gamma) | Vibrionaceae | *Vibrio* | NA | 116 | -6.69359 | 4.0E-04 | Norspermidine biosynthesis, fermentation and carbohydrate degradation |
| Bacteroidota | Crocinitomicaceae | *Fluviicola* | NA | 743 | -2.36485 | 4.5E-02 | None identified |
| Bacteroidota | Chitinophagaceae | Chitino-phagaceae (F) | NA | 46 | -5.84016 | 3.7E-02 | Fermentation and carbohydrate degradation |
| Actinobacteriota | Dietziaceae | *Dietzia* | NA | 509 | -6.64266 | 2.2E-03 | Peptidoglycan synthesis |
| Stramenopiles | Bacillariaceae | *Fragilariopsis* | *F. cylindricus* | 355 | -4.88348 | 8.6E-08 | |
| Stramenopiles | Chaetocerotaceae | Chaetoceros | *C. gelidus* | 956 | -2.34838 | 3.4E-03 | |
| **ASVs differentially abundant in 101 dataset only** | | | | | | | |
| Proteobacteria (Gamma) | Pseudomonadaceae | Pseudomonas | NA | 194 | -22.7872 | 2.41E-13 | Fermentation, carbohydrate degradation |
| Proteobacteria (Gamma) | Pseudo-alteromonadaceae | Pseudo-alteromonas | NA | 149 | -22.2687 | 8.37E-13 | Fermentation, carbohydrate degradation |
| Proteobacteria (Gamma) | Alteromonadaceae | Altero-monadaceae (F) | NA | 133 | -6.52626 | 0.046644 | Fermentation, carbohydrate degradation |
| Chloroplast | Chloroplast | Chloroplast | Chloroplast | 472 | -20.9888 | 1.72E-11 | |


Table A3. The taxonomic assignment, Log2 fold change and abundance characteristics of the eight ASVs that make up the foraminiferal core microbiome.


| ASV | Log2 fold change | Taxonomy | ASV Relative abundance in foraminifera | Total ASV counts in foraminifera | ASV Relative abundance in water | Total ASV counts in water | Forams ASV present in | Samples that ASV is missing from |
|---|---|---|---|---|---|---|---|---|
| ASV956 | -2.348 | Bacillariophyceae (*Chaetoceros gelidus*) | 33.46 % | 783,473 | 2.55% | 54,821 | 27/28 | Fm176b |
| ASV355 | -4.883 | Fragilariopsis (*Fragilariopsis cylindricus*) | 5.30 % | 240,554 | 0.04% | 672 | 24/28 | Fm176b, Fm101a, Fm101b, Fm101e, |
| ASV1447 | NA | Flavobacteriaceae (Family) | 2.40 % | 55,735 | 0.45 % | 9,135 | 23/28 | Fm176b, Fm101d, Fm101e, Fm101g, Fm301a |
| ASV1459 | 2.551 | Pseudoalteromonas | 2.96 % | 91.180 | 3.76 % | 85,466 | 25/28 | Fm176b, Fm101e, Fm301b |
| ASV1392 | 2.277 | Flavobacteriaceae (Family) | 0.72 % | 19,781 | 1.46 % | 29,237 | 24/28 | Fm176b, FmBB2a, FmBB2c, Fm101e, |
| ASV122 | NA | Paraglaciecola | 1.88 % | 61,050 | 0.88% | 21,455 | 25/28 | Fm176b, FM301a, Fm301f, |
| ASV308 | NA | Halieaceae (Family) | 0.88 % | 31,168 | 0.27 % | 5,374 | 26/28 | Fm176b, Fm101e |
| ASV833 | -3.7123 (sig at G and F level) | Bradyrhizobium | 0.09 % | 1799 | 0.004 % | 83 | 23/28 | Fm176b, Fm101b, Fm101c, Fm115a, Fm301d |




## 7 Data availability

The raw 16S metabarcoding dataset was submitted to the NCBI Sequencing Read Archive (BioProject accession PRJNA984332). *N. pachyderma* 18S sequences can be found at NCBI, GenBank Accession numbers OR137988-OR138014. The environmental data, pairwise PERMANOVA summaries, ASV table and sequence files used for this analysis can be found in the supplementary material on Figshare (DOI: 10.6084/m9.figshare.28915598). All code used was standard scripts for each package utilised.

## 8 Author Contribution

CB contributed to the conception and design of the work, gained funding to carry out the molecular work and wrote the manuscript. KD contributed significantly to the writing of the manuscript and conception of the work. AP made a substantial contribution to the conception of the work, the acquisition of funding for sample collection, collected samples in 2017, and contributed to the editing of the manuscript. RT collected samples in 2018 and contributed to the editing of the manuscript.

## 9 Competing interests

The authors declare that they have no conflict of interest.

## 10 Acknowledgements

The authors wish to acknowledge the work of Steve Mitchell at the University of Edinburgh TEM facility for substantial contributions to sample processing and TEM analysis. Sample collection was funded by the Natural Sciences and Engineering Research Council of Canada (NSERC) Discovery Grant (RGPIN-2016-05457) awarded to AJP. Some of the data presented

herein were collected by the Canadian research icebreaker CCGS Amundsen and made available by the Amundsen Science program, which was supported by the Canada Foundation for Innovation and Natural Sciences and Engineering Research Council of Canada. The views expressed in this publication do not necessarily represent the views of Amundsen Science or that of its partners. Transmission Electron Microscopy costs were covered by the Marine Alliance for Science and Technology for Scotland (MASTS) small grant scheme awarded to CB, and molecular lab work was support by a Carnegie Research Incentive Grant awarded to CB.

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
