# Peer review of "The microbiome of the Arctic planktonic foraminifera Neogloboquadrina pachyderma is comprised of fermenting and carbohydrate-degrading bacteria and an intracellular diatom chloroplast store."

_EGUsphere, 2024_

## Author Response (AR1)

A. A summary of revisions to the manuscript due to the new analyses as requested by reviewer one.

1. **Section 2.3** New filtering methods. The new filtering methods have now been incorporated in this section, including filtering of samples based on frequency, taxonomy and contamination. This new dataset contains **1548 ASVs compared to 130ASVs** from the previous analysis, and hence some sections of the manuscript have gone through major revisions. (Note that reanalysis has meant that ASV numbering is completely different from the original manuscript, so reviewers will note these changes throughout.)

2. **Section 2.4 Statistical methods:** New methods for handling and analysing the compositional dataset are highlighted in this section, including centred log ratio transformations and using a robust Aitchison distance dissimilarity for ordination and PERMANOVA on the transformed data as requested by reviewer one. New analyses, PICRUSt2 and ALDEx2, were carried to infer differentially abundant metabolic pathways identified in the foraminiferal microbiome versus the water column. These methods are reported here.

3. **Section 3.2.2 Differential abundance analysis of ASV in foraminifera versus the water column** The DESeq2 analysis was carried out on the dataset which included only forams from where water samples were gained and the seawater samples, as requested by reviewer one, and is reported here.

4. **Section 3.2.3** This is a new section with the results of the PICRUSt2 analyses.

5. **Section 3.3 and subsections within – bacterial ASVs and chloroplast ASVs**. These subsections have had major revisions due to the new larger dataset.

6. **Section 4.2.1 – The core microbiome discussion.** This has been revised, not because the core microbiome has changed from the previous addition, but to add in information regarding PICRUSt2 pathway abundances and the interpretation of the ASVs and their roles. Reviewers rightly requested that analysis of the core microbiome should be more cautious, and this is reflected in amendments made to this section.

7. **Section 4.4.2** is new and incorporates the differentially abundant ASVs between water and foram microbiome and links them to differentially abundant pathways (Picrust).

8. Relative abundance bar charts (now figures 3 and 6) have been updated due to new larger dataset, and TEM figure (now Fig 8 and 9) has been changed in response to reviewer feedback.

B. Response to reviewers' points

Reviewer 1

In their paper, Bird et al. investigate the prokaryotic interactions of the polar foraminifera *N. pachyderma* in Baffin Bay using a combined approach involving 16S metabarcoding and TEM images. While the manuscript offers valuable insights into the core microbiome of *N. pachyderma*, it is important to address the limitations of the dataset and consider repeating some analyses to mitigate potential biases. Moreover, the data analysis strategy requires clearer presentation and justification. Additionally, the conclusion section needs to be included.

We thank reviewer 1 for their thoughtful and constructive feedback.

In the first instance we have addressed the two following comments:

1. At line 177 (Pg. 9), the authors report that the ASV reads were transformed into relative counts and a threshold of 0.1% abundance was used to filter out ASVs. However, amplicon data are inherently compositional, meaning the proportions of the different ASVs are not independent of each other, which can lead to incorrect estimates of similarity between samples. After removing contaminants such as mitochondria and eukaryotes, the read counts should be transformed into centred log-ratios, and the similarity of the samples should then be calculated using the Aitchison distance [1].

Thank you for this input. In our original data analyses, we carried out RA normalisation and removal of 0.1%, and Bray Curtis, in line with previous literature, as cited. However, we very much appreciate reviewer 1's invaluable experience and input here, and in line with their recommendations have carried out the following:

- In filtering ASVs we removed only those that had a total frequency count of 50 across the entire 65 samples including controls. This has replaced the removal of <0.1% after converting to relative abundance.

- Mitochondria and eukaryotic sequences have been removed, and those not assigned beyond the "kingdom" level (as previously performed).

- We used Decontam (R-package) to remove contaminants from our dataset, using the prevalence with batch method. This then gave us our final dataset for analysis.

- We have generated a robust Aitchison distance matrix after converting to robust clr, (rclr). This method works well with sparse data sets, such as our foram microbiomes, (Martino et al 2019).

- Ordination was carried out in q2-gemelli using a RPCA (robust PCA).

These steps have now been clearly documented in the manuscript. Since the number of ASVs retained by this filtering method is far greater than those retained by the previous filtering method, we also re-did all our other analyses.

2. It is unclear which analyses were performed. In line 184 (page 9) the authors generally refer to the mvabund R package and successively mention the generalized linear model but the name of the test is not reported. (Possibly anova.manylm?). Similarly, in line 194 the authors mention the R function adonis for performing a multivariate analysis of variance. This function performs a permutational multivariate analysis of variance (PERMANOVA) and the authors should indicate the number of permutations used in their analysis. Both tests should be repeated using centre-log ratio transformed data and the results should be reported in a table.

The mvabund test used was anova.manyglm, our apologies for this omission. However, we have removed the manyglm analysis form our paper since this was being carried out directly on

counts, and we chose to use the PERMANOVA on the rclr as more appropriate to the compositional dataset. We have now carried out our PERMANOVA (Provenance, Depth and Station) using *Adonis* on the rclr transformed data, as requested. Number of permutations is 999 (the default setting), and the details of this can be found in section 2.4.

Final responses to Reviewer 1:

The authors have carried out the new analyses as suggested by Reviewer 1. This has resulted in a much larger dataset after filtering (1548 ASVs compared to 130 ASVs). As a result, we now address below those comments that remain relevant to the new dataset.

Comments:

1. The authors performed differential expression analyses using DESeq2. However they do not clarify whether they used pre-processed data or the raw read counts as input for the function, as recommended by the algorithm's developers [2], [3].

We carried out DESeq2 analysis on the raw counts as recommended, this has now been clarified in the manuscript. We thank the reviewer for highlighting the omission. The analysis has been re-done given the new dataset. Also, we have carried out this analysis excluding foraminiferal samples that were taken from stations where we have no water data (as suggested by the reviewer in point 2 below). We also re-did the analysis on the station 101 dataset where we have a more robust number of foraminiferal samples, and where we could remove "Station" as a possible factor. **See Section 2.4, lines 182-189**.

2. The results of the permutational multivariate analysis of variance presented on line 273 (page 13) show that location explains 49% of the variability in the ASV profile of *N. pachyderma*. This means that almost half of the observed differences in the microbiome composition among the 28 specimens can be attributed to the different sampling locations. In light of this significant effect of sampling location, the authors should refrain from including the 17 specimens with no contextual seawater samples (Stations 176, BB2, 129, 301) in the differential expression analyses.

We have redone this analysis, using our robust Aitchison clr distances and PERMANOVA, Adonis (qiime2) and to look at the statistical differences of this larger dataset which now contains 1548 rather than 130 ASVs.

We tested the null hypothesis that there was no significant difference in the water samples from the three stations, and also the null hypothesis that there were no differences in the ASV compositions with depth. The results from the PERMANOVA (Adonis) show that station only drives 2.38% of the variation, and that this is non-significant (Pr=0.54). "Depth" however, is the significant driver in differences in the water column.

We tested the null hypothesis that there are no significant differences in the ASV composition between the two provenances in (i) the FW sample set (forams and water from stations 101, 115 and 323) described in the manuscript (**see section 3.2.1**), (ii) the 101 sample set only (due to the more robust numbers of foraminifera for statistical analysis). In both cases a significant

difference was returned. We have tabulated our results for the revised manuscript, as requested by the reviewer – **See Table A1** in the appendix.

3. In lines 261-266, the authors state that a differential expression analysis was repeated with a subset of foraminifera and water samples from station 101 to control for the effect of geographic location, claiming that the results are consistent with those presented in Fig. 4. The results of this control test are reported in Fig. A3 (page 30).

   However, upon comparing the plots, it becomes evident that the results are dissimilar and partially contradict those reported in Fig. 4:

   - The taxa Flavobacteriaceae and Crocinitomix are more expressed in the foraminifera samples in Fig. 4 but more expressed in water samples in Fig. A3.
   - The taxa Chaetoceros, Bradyrhizobium, Moritella, Synedra, and Rubripirellula (among others) are not found in the differential abundance analysis presented in Fig. A3.

   These inconsistencies indicate the significant influence that the selection of samples included in the differential abundance analyses can have on the results.

Some of the confusion here, lies in the agglomerating of ASVs at the Genus level for the figure, and for example the Flavobacteriaceae highlighted are not the same ASVs. This is something we should have raised and thank the reviewer for pointing it out. As mentioned above we have now carried out DESeq2 analysis on two datasets: (i) excluding forams for which there is no water sample to compare, and (ii)  station 101, both with the new much larger dataset of ASVs. Looking at all water stations, there are 13 significantly more abundant ASVs in the foraminifera, and 14 at station 101. Ten of these are the same ASV in both datasets. The results of this are reported in section **3.2.2, lines 265-284, Figure 4 and Table A2**.

4. Given the relevance of the surrounding community in determining interactions with the foraminifera (line 273, page 13), the authors should be more cautious in their interpretation of their results. A thorough ecological discussion on the prokaryotic community in the water column samples, taking into account the environmental parameters measured, should be included given the importance of geographic location before speculating on the potential role of a given taxon as part of the *N. pachyderma* core microbiome.

   For example, the authors could consider including the time since the ice break-up in their analysis to provide ecological context to the prokaryotic community, which is dominated by Bacteroidetes and Proteobacteria in particulate organic matter in sea ice and sinking particles under the ice [4].

As reported under point 2 above, in our new analyses we have shown that "Station" drives 2.3% of the differences in ASV composition (rather than 49%), and is not significant (PERMANOVA, *Adonis*). However, the provenances "foraminifera" and "water column" are significantly different in both the FW and 101 datasets. Here then, we have been able to reject the null hypothesis that

there were no differences in the ASV compositions between "Provenances" and accept the null hypothesis that there are no differences between "Station", indicating Provenance rather than Station (location) has more influence on the ASV composition. We have **changed Section 3.1** of the manuscript to reflect this new data but still agree with the reviewer that we should be cautious in interpreting our results, particularly given the "snap-shot in time" nature of the water samples in particular. In light of this, we have provided a more cautious interpretation of the ASVs that make up the core microbiome (**see lines 465-501**), and interestingly, despite the new total of 1548 ASVs, the core microbiome remains unchanged (except for the loss of one ASV as a contaminant). Indeed, our water column data shows a very significant difference between the water column ASV composition across the water depths, and since we unfortunately do not have depth data for our foraminiferal samples which were collected via a 200m haul, a full discussion of the water column composition at each depth in relation to the foraminiferal microbiome is not possible with this dataset.

Whilst we recognize the importance of geographic location and environmental parameters in shaping these interactions, including an in-depth discussion on the prokaryotic community and factors such as the time since ice break-up would require a significant expansion of our study's scope and objectives. We have aimed to provide a focused analysis on the foraminiferal core microbiome, acknowledging the broader ecological context where relevant. We agree that future studies could benefit from a detailed ecological perspective, and we will certainly consider this in our subsequent research efforts.

We have added a further analysis to our data, using PiCRUSt2 to understand what metabolic pathways the bacterial populations in the foraminiferal microbiome versus the water column, are predicted to be capable of. This can be found in **section 3.2.3 lines 287-301 and Fig. 5.** This analysis allows us to consider the source of the bacteria in the foraminiferal microbiome.

5. Line 435 The authors briefly mention the potential differences in the core microbiome or diet at different ontological stages. Since no data is presented on the size of the specimens analysed this aspect should be discussed.

In light of the new dataset, this statement has been removed.

6. Lines 422-424 (page 20): The authors do not present any evidence beyond the low ASV richness to support the classification of specimens Fm176b and Fm101e as gametogenic. Other potential explanations for this pattern should be discussed.

In response to this and other comments, and to the larger dataset we now have available, this sentence has been removed from the manuscript.

Technical corrections:

7. The datasets, the ASV sequences utilized for the analyses, including the code should be made accessible in public repositories to ensure the reproducibility of the results.

The raw data is publicly available a NCBI (BioProject accession PRJNA984332). The code used for this analysis is not custom code, written by the authors, but standard commands for the R packages used. We have provided all the details of each package utilised where code for all the functions used can be found.

8. Station 60 is mentioned in Table 1 but not plotted in Map in Fig. 1 nor mentioned in the text.

Our apologies, this shall be rectified

9. Lines 243-244 should be moved to the methods section

The authors are a bit unclear about what exactly should be moved within these lines.

10. The conclusion section is missing

Again, our apologies. This is rectified in the updated manuscript.

11. Line 819 correct reference format; Greco et al 2021 is cited in the text but not present in the reference list; Greco et al 2019 reported twice in the reference list

This has been rectified and all references have been checked, thanks to the reviewer for spotting these.

Response to reviewer 2:

**General comments from the Authors**

We thank reviewer 2 for taking time to review our manuscript, and, in particular, for taking great care to flag even small, but important errors. We are very grateful to them. One of reviewer 2's great concerns is the lack of evidence in the TEM imaging for a role of kleptoplasty for diatom chloroplasts in *N. pachyderma*. We want to stress from the start that we are not asserting that the chloroplasts observed in TEM and those identified through 16S metabarcoding, are kleptoplasts in the traditional sense of the word. We have no data to support photosynthesis by the chloroplasts, or of the stability of the chloroplasts in the *N. pachyderma* cell. We had hoped that we had made this very clear in the manuscript, but reading the comments of reviewer 2, we conclude that there is room for an even clearer stance on this, and we will edit the manuscript accordingly.

We believe that this dataset provides some new and important information that should be available to the foraminiferal/plankton scientific community. Given the often-opportunistic nature of sampling in Polar regions, providing this information enables new framing of questions about *N. pachyderma* in Arctic waters, and the right experiments to be carried out to answer them, by whomever has the opportunity to do so.

**Major points:**

- *As written, the rationale for this study is compelling (lines 90-94)—the species is important to the Arctic, which is a rapidly changing habitat due to global warming. That passage is a bit misleading, however, as it notes the data presented here directly impacts understanding of trophic interactions, seasonal shell geochemistry, population dynamics modeling, carbonate flux, and evolutionary pressures. While insights can be gleaned about trophic interactions and – perhaps—evolutionary pressures, the sampling was not done seasonally, so little to nothing can be asserted about carbonate geochemistry, carbonate flux, and population dynamics.*

We accept Reviewer 2's point that our data set does not directly address understanding of seasonality, population and carbon flux, although we do believe it is useful in providing additional data to existing literature. We have therefore amended lines now 94-96 to focus more on trophic interactions and evolutionary adaptations.

However, in terms of the geochemical significance, we believe that if the chloroplasts are in future confirmed to be photosynthesizing inside the foraminifera, their presence will have implications for shell isotopic ratios as observed in species that host photosynthesising symbionts. It is therefore imperative that we communicate this finding so that the scientific community can research the role of these chloroplasts and their impact on the host. Is this an adaptation that has led to the success of this species in the Arctic, and does it impact shell chemistry and therefore affect the palaeorecord?

- *Overall, the contribution is mostly well written and mostly logical. There are, surprisingly, considerable errors in mechanics and grammar, as well as minor points, which are listed separately, below these more-substantial points.*

We thank the reviewer very much for taking the time to point out the grammatical errors, which we have corrected where the sentences remain in place.

- N. pachyderma *is touted as a polar / high-latitude taxon, yet lines 592-593 indicate it also occurs in the tropics. This contradiction should be explained a bit more thoroughly in the context of impact to high (or low) latitudes.*

We have added detail on the biogeography of this morphospecies in the first paragraph of the introduction in lines 1-45

- *It is rather unfortunate that the samples for DNA barcoding were collected in a different year and region from those collected for cellular ultrastructural analysis (TEM). How do the authors truly know that the TEM results (collected in 2018) are representative / comparable to sequencing results from populations collected in 2017?*

We agree that there would have been several advantages to having parallel samples from each sampling station on which metabarcoding could have been carried out alongside TEM. As stressed by the reviewer this would have enabled a strong case to state that the two datasets were representative/comparable with one another. Unfortunately, this is not the dataset we have, given the opportunistic nature of the samples available for these analyses.

To avoid any confusion, we have specifically taken them as stand-alone projects. The TEM images highlight large numbers of chloroplasts within *N. pachyderma*, which to our knowledge has never been reported before. In our manuscript we highlight that the limited preservation does not allow us to unequivocally state that these chloroplasts are from a diatom source. **See lines 413-415, and line 620.**

For the 16S metabarcoding on average 53.3% of ASVs come from chloroplasts and this data therefore supports the observations in the TEM. In 16S metabarcoding we can also state that most chloroplast sequences come from diatom chloroplasts, (44.5% from diatoms specifically) concluding that *N. pachyderma* samples cross Northern Baffin Bay at the time of sampling contain a high proportion of diatom chloroplasts compared to other chloroplasts/bacteria.

One advantage of having samples from different temporal and spatial scales is that we demonstrate, using these two methods, that chloroplasts are found in high numbers in *N. pachyderma* in the summer months across large spatial scales within the Bay and in consecutive years, and it is not therefore just a snapshot in time or space.

- *Given the assertion of phototrophy (line 544), a short passage should discuss light levels at the sample sites and water depths in different parts of the year, with literature citations. Are other N. pachyderma known to photosynthesize?*

In line 544 we are not asserting that *N. pachyderma* is kleptoplastic, but that given the significance of this adaptation to other protists, it is very important to confirm the nature of the chloroplast-foram relationship in this species going forward. We have amended this to more clearly state this distinction and this now sits at **lines 681-685**.

Now that this work has revealed the presence of large numbers of chloroplasts, we are recommending further studies to understand the role of these chloroplasts in the survival of *N. pachyderma*. We know those benthic foram species that eat diatoms, but are not kleptoplastic, do not contain so many chloroplasts in their cytoplasm (**see lines 600-605 for this discussion**) so the question here is why do we see so many? We aim to discuss the various possibilities in this manuscript to enable other scientists who may have opportunity to collect samples from the Arctic, to test these hypotheses.

No planktonic foraminifera have yet been proven to be kleptoplastic, and this would be the only means by which *N. pachyderma* could photosynthesize. We recommend investigation of other *N. pachyderma* genotypes to investigate whether this is an Arctic adaptation, or a more widely held adaptation across different biogeographies/genotypes, as discussed at **lines 625-637**.

- *Lines 87-90 state that 16S rDNA is used here to determine prokaryotic biotic and trophic interactions, and potential symbiotic associations with TEM utilized to note ultrastructural attributes. It is not clear how these methods can be used to inform that these observed chloroplasts are of diatom origin. More specifically, how can ASVs be assigned to diatoms (eukaryotes) from 16S rDNA sequence data (e.g., lines 224-229, Fig. 3)? How were chloroplast ASVs assigned to diatoms (Fig 4, line 252)? While my*

Our TEM images demonstrate chloroplasts, and as mentioned above, in **Section 3.4** (**frustules lines 394-399 and figure 8**, which has been modified to focus on the frustules present). We highlight some diatom chloroplast characteristics that can be identified, and others that unfortunately cannot, given the fixation preservation level achieved here (**lines 414-416**).

16S ASVs can be assigned to chloroplasts from specific genera because the chloroplasts contain 16S genes in their genomes, given their considered evolutionary root as cyanobacteria. These 16S sequences are divergent enough to identify the chloroplast source. Chloroplast 16S sequences from known species, alongside bacterial sequences are part of the SILVA database used to assign taxonomy to the ASVs produced in this study.

**We have added this to line 151**. "PCR was used to amplify the V4 region of the 16S rDNA gene of bacteria and chloroplasts".

- *Why doesn't the ASV noted on line 301 align as Chaetoceros? There are other Chaetoceros in Fig. 5. Why the inconsistency? Why doesn't Fig. 6 show Chaetoceros gelidus while it shows the other common diatom, Fragilariopsis?*

The above is now redundant as these figures have been changed. This chaetoceros ASV is now ASV956 and is labelled consistently throughout the manuscript and figures.

[However to answer the above query: The SILVA database used to assign taxonomy to the ASVs clearly identified some ASVs as *Chaetoceros* but was unable to identify ASV15 beyond the class level. This is likely because ASV15 was more divergent from the *Chaetoceros* sequences in the SILVA database. It is common practice to further investigate key ASVs, such as ASV15, by a BLAST search against additional databases, to confirm or indeed, get better taxonomic resolution of sequence information. We performed this analysis, to confirm ASV15 was a Bacillariophyceae and more specifically could be identified as a *Chaetoceros*.

In Figure 5, the data is based on taxonomic assignment of ASVs against the SILVA database, so ASV15 is grouped alone as a Bacillariophyceae. This therefore gives a clear illustration of the relative abundance of this single ASV. We can make this much clearer in the manuscript and appreciate this being flagged by the reviewer.]

- *Another aspect of the contribution is also confusing in that most of the Introduction, Results, and early Discussion focus on trophic aspects (i.e., food and diet) while the last part of the Discussion focuses on kleptoplasty. It was refreshing to see that TEM was performed for this contribution because it is perhaps the best way to document a putative kleptoplasty. Unfortunately, the TEM images presented were not convincing for a few reasons (noted below). More TEM images are required for review. If the authors can firmly justify this taxon is kleptoplastidic, then the results should be written in this*

*context. If such justification is not possible, then the authors should consider altering those kleptoplasty passages to reflect phagotrophy and digestion. Further, the ultimate conclusion that N. pachyderma is a mixotroph and kleptoplastidic is rather a leap from TEM analysis of specimens collected once in the summer. At the very least, structural documentation should be obtained to show plastid degradation in winter or very early spring (to account for digestion / mixotrophy). While such collections are likely out of the question, the authors should address these shortfalls in the Discussion.*

We are sorry that the reviewer found this format confusing. Our aim was to investigate the feeding behaviour of *N. pachyderma*, and to assess any potential symbionts (TEM can help to assess living microorganisms versus food). Therefore, our focus has been to report the 16S assemblage in the context of food and diet, and finish with a discussion on the chloroplasts and their potential role - one of which could be as (photosynthesising) kleptoplasts.

We believe that we have now more clearly addressed the shortfalls of our TEM fixation preservation in the manuscript (**lines 414-415, and lines 621-622**) since we agree with the reviewer here. Because our data cannot yet confirm kleptoplasty, we have discussed a range of potential roles of the chloroplasts, extensively modifying **section 4.4, starting at line 578.** We first summarise the definition of kleptoplasty, and the roles that benthic foraminiferal kleptoplasts are considered to carry out (both photosynthetic and non-photosynthetic roles) **at lines 578-591**. We then suggest that due to evidence of major evolutionary innovations being linked to photosynthesising kleptoplasts, it is important to examine this relationship further (**Lines 592-597**.) **In section 4.4.1** we provide arguments for and against a kleptoplastic role (both traditional photosynthetic and non-traditional) bringing in information from other studies. We believe this is fully justified, as kleptoplasty is one possible, and significant role that could be played by the chloroplasts. In **section 4.4.2** we expand this discussion to consider alternatives, including the nutritional value of chloroplasts, **in lines 641-648**, and the benefits of storing or very slowly digesting them for overwintering success, bringing in the literature where this strategy has also been suggested for species of benthic foraminifera.

Our intention was not to assert that *N. pachyderma* has photosynthesising kleptoplasts and is therefore mixotrophic. Therefore, we have removed reference to mixotrophy to avoid this confusion.

- *Figure 7 is adequate documentation of potential kleptoplasty ONLY if additional higher-magnification images are included. It is imperative to be able to clearly examine some individual chloroplasts to determine if they are intact or in degraded state. Also, foraminiferal viability is typically determined by status of mitochondria (e.g., LeKieffre et al 2018 Mar. Micropaleo.; Nomaki et al 2016; Bernhard et al 2010), none of which are shown in this figure / contribution. Lipids should not be black, as shown in panel d.*

As per previous responses, the preservation of our foraminiferal samples for TEM was variable and was not of high enough quality to support or refute kleptoplasty. The TEM images are therefore presented only as a means of demonstrating the uniquely large numbers of chloroplasts within the foraminiferal samples, which itself has never been reported in planktonic foraminifera.

We have modified our TEM figures to make clearer what we are and are not asserting with our data. Now we have two figures, **figure 8** showing intact diatoms (including clear chloroplasts) inside the foraminiferal cell, and a second showing empty frustules outside the cell but closely associated with the shell. This figure highlights that *N. pachyderma* ingests intact diatoms, and potentially then ejects the empty frustules. This may also be indicative of the diatom-derived POM that *N. pachyderma* uses as a feeding cyst.

The **second TEM figure, figure 9**, demonstrates the high numbers of chloroplasts found inside the foraminiferal cell. We stress that the limited preservation level does not allow us to observe the numbers of membranes around the chloroplasts for example, but that it absolutely enables us to demonstrate the very high numbers. We compare the numbers in our images with those of both kleptoplastic benthic foraminifera and those that eat algae such as diatoms but are not considered kleptoplastic.

The reviewer should note that the images in the submitted draft are not the highest quality given the nature of the presentation (pasted into word), and that the .png images that will be uploaded with the manuscript will be much clearer.

In our original response to the reviewer, we agreed to remove labelling of the lipids. However, we would like to further push back on this. The lipids in our TEM images are black due to the fixation and staining protocol we have used. Fixing lipids with osmium tetroxide and staining with uranyl acetate mean that the lipids will appear black or very dark in the TEM micrographs. We have added a sentence in the methods at line 214-215 to this effect. We also note that several papers with TEM images of foraminifera, also show abundant black lipid droplets in the presence of sequestered chloroplasts. For example, https://doi.org/10.1093/femsec/fiz046 and https://**doi**.org/**10.1111/1462-2920.14433** .

- *TEM images provided in Fig. A6 are equally too low in magnification to be very helpful in the context of documenting kleptoplasty. It appears that the images in panels f, g, and h have degraded chloroplasts (i.e., evidence of phagotrophy rather than kleptoplasty). Organelles in Panels b and c are difficult to identify at all (very low magnification).*

This figure has been modified slightly to fit the page better, and the figure pasted into the word document is a higher quality version than the previous draft. Nevertheless, we iterate what we have said throughout, these images are to document the abundance of chloroplasts and not their function.

- *More specifically regarding Fig. 7, there is no scale bar in b; the scale in d cannot be correct (caption states bar is 1 micron but that would make those lipids and plastids all far less than 1 um, which is highly atypical; scale in e has no length assigned; panel e should be oriented as it appears in panel c (c has box outlining panel e). Panel b mostly shows the exterior (if one believes the caption); it is not clear why that image was selected as it barely shows N. pachyderma cytology. Fig 7 caption is unclear in places ("pore shape" noted for panel a—presumably intended as "cross section of pore plugs and inner organic lining"?). Further, panel b lacks any "f" label (see caption).*

This figure has now been split into 2, and is now Figure 8 and Figure 9, as described above. The figure legends have been revised accordingly (including the helpful suggestion about labelling of the pore plug), and the scale bars made clearer.

- *Greco et al. (2021) is cited often (e.g., line 61, 82, 85, 396, 413, 443, 457, 481, 533, 588, 612), yet that reference does not appear in Reference list. It is rather unfair to the reviewer and reader that a paper is cited >10 times but details of publication are not given.*

Our sincere apologies for this. It has now been added.

**Reviewer Minor but important points:**

- *The title does not indicate the report is on a planktic foraminifer.*

We have inserted the word "planktonic" into the title.

- *Line 321 is arguable because the two foraminifera could have phagocytosed while at a given water depth and then migrated vertically to another layer / depth horizon.*

The reviewer is correct, *N. pachyderma* migrates through the water column, over its life cycle with juveniles being found at the surface and more mature specimens being found below 50m-200m in general. If the chloroplasts are stored for any length of time therefore, the prey may have been phagocytosed in the upper water column, prior to the foraminifera descending as it matured. If the chloroplasts are rapidly digested however, this would not hold true, as foraminifera do not vertically migrate over diurnal periods of time (e.g. Meilland et al., 2019 https://doi.org/10.1093/plankt/fbz002; Manno & Pavlov 2014, DOI 10.1007/s10750-013-1669-4). We are happy to omit line 321.

- *Please use the proper term for the hard parts of foraminifera: "tests", instead of "shells" which is used throughout the document.*

We chose to use the term "shell" rather than "test" as the word "test" has other meanings in English, and "shell" is used in many published manuscripts. We prefer to keep the term shell.

- *Line 43: please be more specific about "within a short time frame".*

**Now at line 50** we have changed "within a short time" to "ice-free conditions projected to appear between 2030 and 2055" and include additional references to support this; Jahn et al 2024 https://www.nature.com/articles/s43017-023-00515-9 and Kim et al. 2023 https://www.nature.com/articles/s41467-023-38511-8).

- *line 53: omit "habitat".*

Ok.

- *Beginning of sentence spanning lines 71-73 is awkward ("Metabarcoding investigations of other taxa within the Neogloboquadrina genus...").  Why not simply "Microbiome metabarcoding of Neogloboquadrina species..."?*

We agree and have changed this. **Now at line 75**

- *Line 112: a colon should replace the semicolon.*

OK – now at **line 60**

- *Line 116-117: what is "salt adjusted phosphate buffered saline"?  First, "salt adjusted" should likely be hyphenated.  Second, what salt(s) is adjusted and from what concentration to ending concentration?*

**At line 165** now written "TEM buffer (4 % glutaraldehyde, 2 % paraformaldehyde in salt-adjusted phosphate buffered saline (PBS with 24.62g/L NaCl added)),"

Marine or salt-adjusted PBS is modified to match the salinity of the marine environment from which the samples were taken, to prevent osmotic stress and bursting of cells. Salt was added to the PBS to account for the marine conditions. For 35ppt, NaCl is added to PBS at 24.62g/l.

- *Line 142: presumably need to insert comma after parenthetic.*

Ok

- *Line 174: Shouldn't "faith pd" be capitalized to read "Faith PD" or "Faith Phylogenetic Diversity"?*

Yes, now corrected.

- *Line 193 requires comma after assemblages. Omit comma after "differences" in line 195. Line 203: insert comma after parenthetic.*

Corrected.

- *Lines 213-217 largely are not results, but sample site information.  Further, some of the statements lack literature citations.*

Descriptions of sampling stations are found in the methods section 2.1 at **lines 108-113**. Results section 3.1, **lines 218-224** have been edited to provide only the collected data.

- *Line 265: remove comma after "column". Line 330: phylum should be capitalized in both instances. Sentence on Line 355 is irrelevant to TEM results (section topic).*

Suggested changes have been made.

- *Passage on line 390-391 discusses predicted reductions in non-spinose foraminifera yet the reader has not been informed if N. pachyderma is spinose or non-spinose.*

This is now in first line of introduction – **line 34**

- *Line 416: What is a detrital diatom?*

Diatoms that are dead and part of the detritus. At line 399 "dead" has been added: "detrital (dead) diatoms"

- *Why are bacteria necessarily linked to the diatoms (lines ~415-418)? Bacteria and archaea can live independently from diatoms.*

Yes, we agree that bacteria and archaea can live independently from diatoms. This sentence relates to the discussion at lines 472-481 on the bacteria, and in particular the flavobacteria found in the diatom phycosphere, and their greater relative abundance in the foram than in the water column. This is hypothesised to be a function of the large numbers of flavobacteria feeding in the phycosphere, so as *N. pachyderma* eats the diatom, it also consumes these bacteria disproportionally to other species not associated with the phycosphere. It does not exclude the idea that bacteria could also be taken up directly from the water column and from other kinds of POM.

This part of the manuscript has been extensively re-written. **Section 4.2.1 and lines 470-485** discuss the bacteria that are known diatom associates. The Flavobacteria and Pseudoalteromonas identified in the core microbiome, are groups that are shown to co-occur with diatoms, and indeed Psuedoalteromonas are known to be first colonisers of degrading diatoms. Both grow on diatom exudates, and so the evidence suggests that these bacteria are likely to have been phagosytosed with the diatom food source.

- *Line 425-426 notes twice that there are eight core ASVs in microbiome yet Table 2 lists nine "core" ASVs in microbiome (see also line 433). Why the difference? Which is correct?*

This has been corrected and updated because of the new analysis. **See Figure 7.**

- *It is not clear how some specimens were denoted "potentially gametogenic" (lines 462, 525-526). No data is presented on this (not in tables, etc).*

This has been removed in the major revisions resulting from the new analysis.

- *Discussion of light use by foraminifera appears on lines 487 to 488 yet light levels are not reported in the data presented here.*

Light use is not discussed in relation to the foraminifer, but in relation to ASV74 (now ASV308) which is assigned to the order Cellvibrionales many of which can utilise light in bacterial photosynthesis/energy generation. Since our *N. pachyderma* samples were collected from the photic zone (above 200m), it is likely that some of the bacteria they ingest are photosynthetic. ASV308 is a core member of the microbiome and therefore was discussed in that context, as likely to be photosynthetic, or using rhodopsin for energy generation. **We have attempted to make this clear at lines 487-491.**

- *Line 493 is oddly worded—how can microbes be clustered in a gene?*

This has been improved **at line 499**.

- *The passage from Line ~493 to 497 is confusing because it seems to be advocating that these foram populations are $N_2$-fixing but that was not measured. Further, it is not clear that "our findings" on line 497 are supported by Fernandez-Mendez et al. (2016).*

This is a discussion on the metabolism of the bacteria that are taxonomically assigned to particular ASVs, and not on the metabolism of the foraminifera. ASV116 is assigned to the nitrogen fixing *Bradyrhizobium*, and it is a core microbiome constituent. We highlight here, that our data on the presence of these organisms agrees with previous data showing their abundance in the region compared to other *nifH* types. This has been reworded for clarity at **lines 498-502**

- *Authors must explain why ASV27 is called Aurantivirga (line 498 and perhaps elsewhere) yet lines 503 to 505 says it is a Tenacibaculum species. Which is the reader to believe? Then the authors pontificate about fish teeth and how this ASV27 provides extra calcium to the foraminifer for calcification, which is pure speculation.*

This has been reworked, as the reviewer had clearly misunderstood what we were trying to say. **Lines 476-482** discuss the presence of two *Flavobacteriaceae* ASVs in the core microbiome. An NCBI BLAST search further identified one of these two ASVs as belonging to the genus *Tenacibaculum*. These bacteria require elevated calcium for growth and are therefore often found associated with fish teeth. This is of interest since the foram shell is made of calcium carbonate and therefore may act as another high calcium niche that these bacteria inhabit. We thought it an interesting small detail. **See lines 483-486**.

- *Line 537: please explain what "intracellular ingestion" is. Ingestion is the act of taking something from outside the cell / body into the cell / body. Most biologists would simply say the foram phagocytoses the complete diatom (frustule + cell).*

We are trying to highlight that *N. pachyderma* phagocytoses the complete diatom and then digests the cell (but arguably not the chloroplast straight away) as opposed to many benthic foraminifera which crack open the frustule externally and phagocytose the soft cell material leaving the frustule on the outside. The term "intracellular ingestion" was used by Pinko et al 2023 to describe this, (cited in the manuscript) but we have now reworded it at **line 575**.

- *While the term kleptoplasty was used three times in the Abstract and once earlier in the contribution (line 440), it is not until section 4.4 that "kleptoplastic behaviour" is asserted for N. pachyderma. The term is never fully defined although a partial definition appears on line ~544.*

Again, we wish to make clear that we are not asserting the *N. pachyderma* is kleptoplastic. But the reviewer makes an excellent point that we have not defined it.

Kleptoplasty has now been fully defined in Section 4.4 where it is most appropriate to do so. **See lines 582-585.**

- *Provide references for foraminifera, ciliates and dinoflagellates that are kleptoplastic (line 543).*

This is inserted at **lines 581-582**.

- *Proper spelling is "byproduct" line 549, vs "biproduct".*

Apologies. This is corrected.

- *The citation of LeKieffre et al. (2018) on line 554 should be the LeKieffre et al. 2018 paper in Marine Micropaleontolgy, not the one in Mar. Biol. The citation on line 583 is properly cited as the 2018 Mar. Biol. paper.*

Due to revisions in the manuscript, only the marine micropaleontology paper is now cited.

- *Powers et al. (2022, Frontiers in Mar. Sci.) should be cited on line 575, with Gomaa et al., as these two papers describe some of the functions of the N. stella kleptoplasts via gene expression (transcriptomics).*

Ok.

- *Regarding mixotrophy and kleptoplasty in foraminifera (line 583), I believe Cedhagen (1991) documented this long ago for Nonionellina labradorica. That paper should be cited somewhere in paragraph spanning lines 598 to 604.*

We agree, and it is now cited here **– line 588**

- *Authors are reminded that many (perhaps most) forams do not have carbonate tests (shells), so that should be noted in passage about geochemistry (~ line 625).*

We have added the word "calcifying" to **line 669**

- *What exactly is "metabolic C" (line 628)? An internet search reveals it is a type of health supplement of Vitamin C.*

Our apologies that this is indeed unclear. C is carbon. In this context the metabolic carbon is that which has been produced in respiration inside the cell (by endobionts) and so has an offset C-isotopic ratio that can be reflected in the shell C isotopic ratios. We have added "(respired)" **to line 672.**

- *Line 630: The phrase "sediment assemblage" typically refers to the benthos (organisms that live on or in the seafloor). Perhaps authors intend "fossil record" or "sedimentological record" instead?*

Yes, we do. We have amended to "fossil record" **at line 674**

- *Line 633 to 634 should add "calcareous" to read "...throughout the lifetime of calcareous foraminifera (Spindler...".*

Added "calcareous" now at **line 677**

- *Statement on line 641 should change as photosynthesis was not shown, only inferred.*

This sentence has been reworded to reflect that. We are not claiming that we have proven photosynthetic kleptoplasty. **See lines 683-685**.

- *Line 731 should be italicized (see same genus on prior page)*.

Thanks for spotting this.

**Mechanics (also important):**
- *The tense of the document is a mix of past and present, often in the same sentence; most scientific papers are in past tense. Please change lines 17 to "used", line 19 to "was", line 20 "consisted of", lines 217 - 219 should be past tense, etc.*

We have checked through the document to give consistency.

- *Readers who are red-green color blind will be significantly challenged while reading Figures 3, 4, 5, A1, A2, A3, and A4. The authors should consider changing their color palettes, datum point shapes, etc.*

The colours have been specifically chosen using the Coblis-Colour blindness Simulator as recommended in the submission portal. According to this simulator the chosen colours are suitable for a range of colour vision deficiencies including red-green.

- *More reference strings should begin with "e.g.," because the cited papers are only examples and not an exhaustive list: lines 52, 440, 557, 634. For line 557-558, it is surprising that Bernhard and Bowser 1999 Earth Sci. Reviews is not cited as it compiled all known foraminifera kleptoplasty cases to that time.*

We agree, the number of references would be enormous if we used exhaustive lists. We will add e.g., and we agree that Bernhard and Bowser should certainly be cited and have added this.

- *Remove colloquial wording (e.g., line 49 "...will find itself spatially... should be "will be spatially..."; line 171: "...was carried out..." should be "..was executed..." or similar; lines 272-273 "a degree of clustering" should read "shows clustering..."; line 305 "six out of the seven" should be "six of seven"; line 464: "break down" should read "degrade"; line 466: "carry out" should read " perform".  Line 669 to 670 is obtuse ("There is a degree of clustering...").*
- *Lines 115, 772: Ship names are always italicized.*
- *For Figure 1: suggest using different color or shape for symbols in the two different sampling years.  Resolution will be, presumably, better than that provided for review.*

We have addressed all these points.

- *Methods seem a bit too detailed (e.g., passage from lines 143-148).*

We are happy to go through the methods and make them more concise. However, in the example given here, the PCR amplifications were rather different from protocols used elsewhere, a reflection on PCR requirements for forams stored in RNALater, and since this protocol is not available elsewhere, we have included it.

- *Please fix font style and size ("t" in "taxa" on, e.g., lines 279, 329; 521).*
- *Line 473 "sp." should not be italicized, ever.*
- *Suggest authors comb the text for errors in punctuation, etc. For example, see lines 512, 551, 561 (remove comma before parenthetic).*

- *The order of the key in Figs. A1 and A2 is peculiar because it goes from 0 to 100, to 150 to 200 to 50. Most would arrange from 0 to 50 to 100 to 150 to 200. The gray background in those figures is inappropriate (waste of ink / electrons).*

We have addressed all these points.

- *The Conclusions section is blank / non-existent.*

Our sincere apologies for this. This is rectified.

- *References were not proofread by authors. Issues are (a) at least 46 species names have not been italicized; (b) at least six entries lack the journal name; (c) some entries are gibberish (e.g., Bell & Mitchell, Brummer et al – see penultimate author name); (d) lack of URL for software packages (Lahti); (e) lack of publisher in at least 2 references (Eegeesiak et al., Schiebel and Hemleben); (f) font color is inconsistent; (g) missing spaces between words, (h) inconsistent presentation of journals (some names abbreviated, others not), (i) lack of superscripting isotopes, (j) inconsistent line spacing.*

Our apologies for this. We did proofread our document but acknowledge that there are regrettably some errors. We used a new citation tool for this manuscript, and it is clear we had some teething trouble. We have now amended our reference list.

---

## Referee Report (RR1)

The authors have strengthened their analyses by introducing new filtering criteria and refining their methodological approach. Notable improvements include the application of transformation-centered log-ratio transformations and the Aitchison distance dissimilarity measure for ordination and PERMANOVA. Additionally, the study now integrates complementary analytical tools such as PICRUSt2 and ALDEx2, allowing for a more comprehensive inference of differentially abundant metabolic pathways within the foraminiferal microbiome in relation to the surrounding water column.

The results are clearly presented, and the methodology is well-documented. These refinements contribute to a more robust and informative study.

However, addressing the following points would further enhance the clarity, coherence, and reproducibility of the manuscript.

SPECIFIC COMMENTS:

On page 8, lines 165–168, the authors mention performing rarefaction and calculating alpha diversity metrics. However, corresponding rarefaction curves do not appear to be included in the supplementary material. Adding these curves is necessary as it would help assess whether the sampling effort was sufficient to capture the diversity of the prokaryotic community in both foraminifera and water samples.

In their response, the authors state:

> *"As reported under point 2 above, in our new analyses we have shown that "Station" drives 2.3% of the differences in ASV composition (rather than 49%), and is not significant (PERMANOVA, Adonis). However, the provenances "foraminifera" and "water column" are significantly different in both the FW and 101 datasets. Here then, we have been able to reject the null hypothesis that 4there were no differences in the ASV compositions between "Provenances" and accept the null hypothesis that there are no differences between "Station", indicating Provenance rather than Station (location) has more influence on the ASV composition"*

However, the multivariate analysis performed on the 'Forams only' subset indicates that 48.3% of the variation in foraminifera microbiome composition is attributed to station (lines 315-318, p. 16). In contrast, the same analysis on water samples (stations 101, 115, and 323) does not show a significant relationship with station (Fig. A1). Furthermore, when the 'FW' subset (including both water samples and foraminifera) is analyzed, station is no longer a significant factor. Based on these findings, the authors conclude that 'provenance' rather than location is the main factor shaping the foraminifera microbiome. However, these results appear to be somewhat contradictory. If station explains a substantial proportion of variation in the 'Forams only' subset but not in the water samples or the combined 'FW' subset, additional clarification/discussion is needed in the manuscript. To further investigate this discrepancy, I recommend adding another subset and repeating the analysis using only the foraminifera microbiome data from the same stations as the water samples (which likely corresponds to the 'FW' subset minus the water samples). This additional analysis could help determine whether the observed patterns in the 'Forams only' subset are driven by differences in water column composition, ultimately clarifying the influence of microbial community assembly processes in the surrounding water on the foraminifera microbiome. Given that the authors define the core microbiome using all foraminifera data, it is crucial to discuss the ecological and biological factors shaping its composition to ensure a comprehensive interpretation of the results.

The results support the previously hypothesized POM feeding mode and suggest that *N. pachyderma* also preys on living diatoms.

A recent study by Meilland et al. provides experimental evidence for this feeding behavior in cultured *N. pachyderma* from high latitudes (Type I). Citing this reference and incorporating its findings into the discussion would provide a more comprehensive context for interpreting the feeding ecology of *N. pachyderma*.

*Meilland, J., Siccha, M., Morard, R., & Kucera, M. Continuous reproduction of planktonic foraminifera in laboratory culture. Journal of Eukaryotic Microbiology, e13022. https://doi.org/10.1111/jeu.13022*

Throughout the manuscript, microbiome community structure in foraminifera and water samples is described using expressions such as "% of all ASVs" (e.g., line 329, p. 16) and "contained >50% chloroplast ASVs" (e.g., line 359, p. 18). These phrases describe the number of ASVs rather than their relative abundance (i.e., read proportions) but are referenced alongside figures that likely represent read proportions.

To avoid potential misinterpretation, I recommend explicitly clarifying that these values refer to read proportions and updating the y-axis labels in Figures 3 and 6 to specify "% reads."

The statement regarding raw sequence data (line 248, p. 12) should be moved to the "Data Availability" section. Additionally, to enhance reproducibility, I strongly encourage making the environmental contextual data, ASV table, and sequence files publicly accessible, as previously suggested.

---

## Author Response (AR2)

RESPONSE TO REVIEWER 1

The authors have strengthened their analyses by introducing new filtering criteria and refining their methodological approach. Notable improvements include the application of transformation-centred log-ratio transformations and the Aitchison distance dissimilarity measure for ordination and PERMANOVA. Additionally, the study now integrates complementary analytical tools such as PICRUSt2 and ALDEx2, allowing for a more comprehensive inference of differentially abundant metabolic pathways within the foraminiferal microbiome in relation to the surrounding water column.

The results are clearly presented, and the methodology is well-documented. These refinements contribute to a more robust and informative study.

However, addressing the following points would further enhance the clarity, coherence, and reproducibility of the manuscript.

SPECIFIC COMMENTS:

1. On page 8, lines 165–168, the authors mention performing rarefaction and calculating alpha diversity metrics. However, corresponding rarefaction curves do not appear to be included in the supplementary material. Adding these curves is necessary as it would help assess whether the sampling effort was sufficient to capture the diversity of the prokaryotic community in both foraminifera and water samples.

Response: These curves (Shannon, observed-features, Faith PD) are now added to the appendix (Figure A1).

2. In their response, the authors state:

*"As reported under point 2 above, in our new analyses we have shown that "Station" drives 2.3% of the differences in ASV composition (rather than 49%), and is not significant (PERMANOVA, Adonis). However, the provenances "foraminifera" and "water column" are significantly different in both the FW and 101 datasets. Here then, we have been able to reject the null hypothesis that there were no differences in the ASV compositions between "Provenances" and accept the null hypothesis that there are no differences between "Station", indicating Provenance rather than Station (location) has more influence on the ASV composition"*

However, the multivariate analysis performed on the 'Forams only' subset indicates that 48.3% of the variation in foraminifera microbiome composition is attributed to station (lines 315-318, p. 16). In contrast, the same analysis on water samples (stations 101, 115, and 323) does not show a significant relationship with station (Fig. A1). Furthermore, when the 'FW' subset (including both water samples and foraminifera) is analyzed, station is no longer a significant factor.

Based on these findings, the authors conclude that 'provenance' rather than location is the main factor shaping the foraminifera microbiome. However, these results appear to be somewhat contradictory. If station explains a substantial proportion of variation in the 'Forams only' subset but not in the water samples or the combined 'FW' subset, additional clarification/discussion is needed in the manuscript.

To further investigate this discrepancy, I recommend adding another subset and repeating the analysis using only the foraminifera microbiome data from the same stations as the water samples (which likely corresponds to the 'FW' subset minus the water samples). This additional analysis could help determine whether the observed patterns in the 'Forams only' subset are driven by differences in water column composition, ultimately clarifying the influence of microbial community assembly processes in the surrounding water on the foraminifera microbiome.

Given that the authors define the core microbiome using all foraminifera data, it is crucial to discuss the ecological and biological factors shaping its composition to ensure a comprehensive interpretation of the results.

Response: We have done a series of additional analyses to tease these out. The additional/improved methods are reported in Section 2.4. These include:

- Using adonis2 from the Vegan package as an improved version of Adonis in QIIME2 which better handles unequal sample sizes.
- Using the R package *pariwise.Adonis* to investigate which Stations were driving the statistical differences observed in the foraminiferal dataset
- Testing for homogeneity of multivariate dispersions among groups using bestdisper on vegan to test whether the assumptions of PERMANOVA are violated.
- Implementing Wilcoxon test in ALDEx2 where PERMANOVA assumptions are violated.

The additional tests are reported in a new section "3.2.1 Station and Depth as factors influencing microbial assemblages"

- In the water samples because Depth was significant but Station was not, we considered that the multiple depths within each sample might be masking the effect of station. We therefore carried out a test on 50m water samples only.
- Because station was significant for foraminifera despite not being significant for water, we did several things.
    - First we took a more statistically robust set of foraminiferal samples excluding stations where n=2. Station remained significant in this dataset.
    - We also did pairwise analysis on both the full and the robust datasets to identify which pairs of stations were significantly different.
    - As suggested by the reviewer we made a small dataset including only foraminiferal samples from the stations where water was also taken so we could directly compare Station significance on the water and the forams independently. This small dataset violated the assumption of homogenous dispersions for PERMANOVA and therefore we ran ALDEx2 Wilcoxon tests as mentioned above.

We have discussed the results in a new brief section "4.1 The influence of station and depth on microbial assemblages". The main conclusions are

- Depth is the strongest driver of differences in microbial assemblages
- Only a small subset (4/21 pairs of stations in the foraminiferal dataset) are significantly different and these drive the global PERMANOVA results.
- Station has a more subtle influence than Depth, and those stations furthest apart are those that show significantly different foraminiferal ASV composition.

- The co-sampled (water and foram) stations, 101, 115 and 323, are not significantly different by any test.
- Therefore, the difference in significance between the foraminiferal and water column datasets likely reflects the broader geographic coverage of the former, along with host-mediated retention of a stable core microbiome through selective feeding.

3. The results support the previously hypothesized POM feeding mode and suggest that *N. pachyderma* also preys on living diatoms.

A recent study by Meilland et al. provides experimental evidence for this feeding behavior in cultured *N. pachyderma* from high latitudes (Type I). Citing this reference and incorporating its findings into the discussion would provide a more comprehensive context for interpreting the feeding ecology of *N. pachyderma*.

*Meilland, J., Siccha, M., Morard, R., & Kucera, M. Continuous reproduction of planktonic foraminifera in laboratory culture. Journal of Eukaryotic Microbiology, e13022.* https://doi.org/10.1111/jeu.13022

Response: Great idea, thank you. At lines 529-534 we have added this reference and some discussion on the species that best support growth in culture versus those identified in the natural environment.

4. Throughout the manuscript, microbiome community structure in foraminifera and water samples is described using expressions such as "% of all ASVs" (e.g., line 329, p. 16) and "contained >50% chloroplast ASVs" (e.g., line 359, p. 18). These phrases describe the number of ASVs rather than their relative abundance (i.e., read proportions) but are referenced alongside figures that likely represent read proportions.

To avoid potential misinterpretation, I recommend explicitly clarifying that these values refer to read proportions and updating the y-axis labels in Figures 3 and 6 to specify "% reads."

Response: This is a good point and does require clarity. We have added a line at lines 176- 179 in the methods to explain that the ASV counts were converted to relative abundances (percentage reads) for the analysis to make figures 3 and 6 and that all percentages discussed are relative abundances (read proportions) converted from read counts. We have changed the y-axis labels to "% Reads" it the figures as requested.

5. The statement regarding raw sequence data (line 248, p. 12) should be moved to the "Data Availability" section. Additionally, to enhance reproducibility, I strongly encourage making the environmental contextual data, ASV table, and sequence files publicly accessible, as previously suggested.

Response: The statement at line 248 has now been moved to the data availability section. A DOI giving access to the CTD data, pairwise analysis, ASV table and sequence data on Figshare (DOI: 10.6084/m9.figshare.28915598e) is also now provided in this section. At time of writing, this DOI has been reserved and will be released when the dataset is published.

RESPONSE TO REVIEWER 2

1. The revision of the manuscript by Bird et al. is quite improved over the original. The revised contribution, with updated figures and Conclusion paragraph, is more focused and clearer, although I am not a specialist in molecular approaches/methods. As with all manuscripts, there remain a few points of concern to consider.

Response: Thanks go to both our reviewers for giving generously of their time and their steely determination to improve this manuscript considerably.

2. The Abstract (line 17) and Introduction (line 70) both imply this species was studied by these authors "throughout the annual cycle of the Arctic", but they only sampled in summer.

Response: Thank you for pointing this out. It certainly was not our intention to imply this, so we have rewritten these lines to make clearer that our work covers only the summer season. In the abstract we indicated that we studied only the summer populations at line 20, but have moved this statement earlier (line 17), to make this clear immediately. At line 70, since this is a broad introductory section, we have just tweaked the wording a little to make it clearer that this is something that will need to be done, rather than something we are reporting. So it now says "To model the impending environmental consequences for this important high latitude species going forward it will be vital to investigate the modern ocean community structure throughout the annual cycle of the Arctic to understand the inter-dependencies of *N. pachyderma*."

3. The statement on line 345 about ASV956 being the most abundant chloroplast ASV and assignment to class Bacillariophyceae is not supported by Fig. 6 in that the dominant taxon shown in Fig. 6 is Chaetoceros (dark green), not Bacillariophycaea (light green). I understand Chaetoceros is a genus of Bacillariophycaea. The authors should check their wording to be sure it corresponds to the figure. If the figure and text are congruent, the authors are urged to check the passage (lines 344-347) for clarity.

Response: Lines 395-387 have been reworded to make this clearer. Since ASV956 was identified to the species level it was grouped in the *Chaetoceros* genus in Fig 6 and so makes up the dark green bar.

4. The authors insist on use of the term "shell" vs "test" (response page 12), yet they use the term "test" to describe the foraminiferal hard part on line 486 and perhaps elsewhere. Why the inconsistency? The external cytoplasm (which is why it is called a test, not a shell) is discussed by them on line 397. Further, a google search of "What is the hard part of a foraminifera called?" results in the term "test", not "shell". The authors can use an incorrect term but let the record show that I strongly object to such terminology.

Response: The reviewer makes a compelling and sensible argument, so we have changed the term shell to test throughout.

5. Also regarding terminology, the authors use an odd abbreviation for foraminifer ("fm"; Table 1, labels in Figs. 3 & 6, etc). If the authors are so concerned about misinterpretations of the term "test" (response document pg 12) then why use "fm", which to most oceanographers, marine scientists, and English-speaking people means fathoms?

Response: This had not occurred to us! We request to leave the abbreviation Fm in place at this time, as it is deeply embedded in the data set from the sequence identities in the original files generated by the sequencer and within all the code throughout the analysis. It would be a massive challenge to change it at this stage, and we hope the reviewer will be content to allow its use on this occasion. We will not use it again.

5. Unfortunately, there are some statements in the authors' response document that simply are not true. The response notes they would add "e.g.," to instances where the cited papers are examples (see page 18 of response document) versus cases where a complete list of existing publications is cited. The "e.g.," appears a few times (lines 38, 48, 72, 74, 636) but there are many more where it should appear, including lines 59, 555, 580-581, 586, 608, 611, 619, 670, 672, 678, and 685, at a minimum. The authors' response also noted (pg 18) they would cite the benthic foraminiferal kleptoplasty review by Bernhard and Bowser (1999), but they did not. A good place to cite it is line 555 and/or lines 608-609.

Response: Please forgive these oversights. We have amended our "e.g." additions accordingly which can be seen throughout the newly marked up submission. We have finally added reference to the paper by Bernhard and Bowser 1999 at lines 646 and 681. We are sorry for this omission - we were convinced we had added this paper in the last round of edits as it is very important for the discussion on kleptoplasty and was one of the first to show how abundant kleptoplasts can be.

6. There continue to be at least a few missing references (aside from not adding the one noted above), including Salonen et al. 2021 (cited on line 647) and Westgard et al. 2023 (cited on line 667). Clearly one of the authors must do a careful and complete check for such omissions through the entire manuscript.

Response: The authors did do a careful and rather painstaking check through the references, including checking the journal abbreviation consistency etc. We are sorry that some references still slipped through the gaps and have checked again. As well as adding a couple of additional references for additional R packages used in the stats analysis and discussion added as requested by reviewer 1, we have added two further references that we missed previously (Quast, and Qu).

7. Regarding TEM images of lipids, I must reiterate that the lipids in Fig. 9a are extremely dark compared to lipids of most benthic foraminifers. For excellent examples of what benthic foram lipids should look like, see LeKieffre et al. 2018 Mar. Micropaleo., Figs. 7 C, F; Fig 10 D, E; Fig. 12 D. The issue is that "electron opaque bodies" that truly are black (electron opaque) exist in benthic forams and the authors should be aware that these differ from lipids. Why are there no (black) lipids in Fig. 9B (no black structures)? Such dark lipids do not occur in most of the images in Fig A6 either. None of these comments on lipids require any changes to the contribution; the comments are only to provide the authors with information.

Response: Thank you for the clarification on "electron opaque bodies" this is useful to understand, and we have modified the manuscript at lines 468-469 and in the Figure 9 legend at line 473, to take this possibility into account, as we feel that it is important based on the reviewers feedback.

**Minor points**

"Microbiome" should be defined at first use in the Introduction. Now, it is defined in the Discussion (line 429).

 Response: This has now been added at line 76.

The proper term for localized "extinction" is "extirpation" (line 61).

 Response: OK, we have changed this.

Define "TEM" (line 78).

 Done

Remove the blank line (line 84).

 Done

Define "interactome" (line 90). Is its use valid in this work? Interactions were not investigated.

 Response: This is the term used by Greco et al., 2021 which is cited here, but we have changed it to 18S microbiome, as Greco et al are referring to the results of the 18S metabarcoding dataset, so microbiome seems appropriate and we agree with the reviewer that interactomes is less appropriate as it usually refers to molecules.

Consider changing the antiquated, generally disfavored "prokaryote", e.g., on lines 92, 319.

 Response: Good point. This has been changed to "bacterial and archaeal" to reflect the targeting of these two very different groups.

Lines 114, 376 and elsewhere: do not start a sentence with a number or abbreviation.

 Response: This has been checked and changed as necessary. However, there are several sentences that start with PCR and with ASV, and we feel that these two abbreviations should be left as such, even at the start of sentences just for ease of reading. There are plenty of manuscript examples where both PCR and ASV have been used at the start of sentences. I am happy to change these, however, if the editor feels it is necessary.

Change "samples" to "sample" on line 254.

 Done.

Add comma after "ASV" on line 284.

 Done

Change "Chloroplasts" to the singular in header to section 3.3.2 (line 326).

 Done.

Omit the errant "d" near beginning of line 372; add comma after Fragilariopsis (before Synedra).

Response: This is not an errant "d" -it is part of the sentence: "Except for Fm176b which had only 3 chloroplast ASVs, Fm176a, c, and d contained a much higher relative proportion of…" so we have made this more clear by stating "Fm176a, 176c and 176d"

The passage on lines 398-400 belongs more in the Discussion than in the Results.

Response: The sentence referred to is : "These observations support the previous literature indicating that *N. pachyderma* consumes diatoms and given the intact nature of the diatoms observed (Fig. 8a), it is not only detrital (dead) diatoms that are consumed, as reported by Greco et al. (2021)."

It has been deleted from this location and rejigged into line 426-427 of the discussion.

Omit comma after "al." on line 441.

Done.

Omit comma after "diatoms" on line 442.

Done.

Line 445 says "sits", which is colloquial and anthropocentric. Rewrite. Consider "is situated".

Response: We have changed "sits" for "remains"

Shouldn't the two "or" on lines 468-469 be "and/or"?

Response: Yes- that makes more sense -thank you.

Check the math for statement on lines 471-472. Should the average be 0.89% instead of 8.93%?

Response: No, this is correct -so we have amended the wording to state "Six bacterial ASVs each contributed small percentages between 0.09 %- 2.96 %, averaging a total of just 8.93% of the core microbiome between them.

Is it proper to report values to the hundredths (lines 471-472, 552, and elsewhere)?

Response: We have chosen to report two decimal places where the numbers are very small (such as in the core microbiome) to distinguish between the ASVs at these lower levels of relative abundance.

Omit comma after "Cellvibrionales" on line 488.

Done

Omit comma after "C" on line 496.

Done

Remove the third "the" on line 547 (to read "...beginning of vertebrae evolution...")

Done

Add comma after "diversa" on line 575.

Done

Line 598 is underlined but other sections of that stature are not underlined. Be consistent.

Done

Capitalize the "k" in LeKieffre (line 605 and, perhaps, elsewhere).

Done

Omit comma after "cytosol" on line 619.

Done

Add citation at the end of sentence ending on line 627.

Done

Omit comma after "paleoenvironments" on line 668. Why use the US spelling given European spelling / words are used elsewhere (e.g., "anticlockwise" in figure caption)?

Response: Good point, spelling has been amended, and comma removed.

Add space in proper location on line 696 ("of2017").

Response: There is already a space here -although it doesn't look like there is!

Suggest changing "this" to "that" on line 697, to read ...microbiome at that time."

Response: Good idea, done.

Hyphenate "nutrient rich" (line 698).

Done.

The supplemental tables continue to use different fonts (see Table A1 vs Table A2).

Response: They are now consistent in Times New Roman size 9.